

# Different roles of water in secondary organic aerosol formation from toluene and isoprene

Long Jia[1,2], Yongfu Xu[1,2]

[1]State Key Laboratory of Atmospheric Boundary Layer Physics and Atmospheric Chemistry, Institute of Atmospheric Physics, Chinese Academy of Sciences, Beijing 100029, China,

[2]Department of Atmospheric Chemistry and Environmental Sciences, College of Earth Sciences, University of Chinese Academy of Sciences, Beijing 100049, China

*Corresponding to*: Yongfu Xu (xyf@mail.iap.ac.cn)

**Abstract.** Roles of water in the formation of secondary organic aerosol (SOA) from the irradiations of toluene-$NO_2$ and isoprene-$NO_2$ were investigated in a smog chamber. Experimental results show that the yield of SOA from toluene almost doubled as relative humidity increased from 5% to 85%, whereas the yield of SOA from isoprene under humid conditions decreased by 2.6 times as compared to that under dry conditions. The distinct difference of RH effects on SOA formation from toluene and isoprene is well explained with our experiments and model simulations. The increased SOA from humid toluene-$NO_2$ irradiations is mainly contributed by O−H-containing products such as polyalcohols formed from aqueous reactions. The major chemical components of SOA in isoprene-$NO_2$ irradiations are oligomers formed from the gas phase. SOA formation from isoprene-$NO_2$ irradiations is controlled by stable Criegee intermediate (SCI) that is greatly influenced by water. As a result, high RH can obstruct the oligomerization reaction of SCI to form SOA.

## 1 Introduction

Water is an important environmental factor that can influence the formation of secondary organic aerosol (SOA) through the physical or chemical processes, and is often represented with relative humidity (RH) or liquid water content (LWC). Toluene and isoprene are two important precursors of SOA, which are representatives of volatile organic compounds (VOC) from anthropogenic and biogenic sources. Both toluene and isoprene can produce glyoxal during their oxidation processes in the atmosphere. As widely reported, glyoxal is a typical precursor of SOA formed in the aqueous phase (Volkamer et al., 2009;



Lim et al., 2010,2013; Ervens et al., 2011; Shen et al., 2016). The difference is that toluene contains an aromatic ring, which

is mainly oxidized by OH radicals, while isoprene contains two C＝C bonds, which can also be oxidized by $O_3$ in addition to

OH. The Criegee intermediates are distinctive species in the ozonolysis of alkenes. Stabilized Criegee intermediates (SCI) can

react with carbonyl products, alcohols, $H_2O$ and organic peroxy radicals ($RO_2$) (Neeb et al., 1998; Calvert et al., 2000; Sadezky

et al., 2008; Huang et al., 2013; Chao et al., 2015). SCI-derived reaction products have been observed in the aerosol phase

(Sadezky et al., 2008; Huang et al., 2013; Sakamoto et al., 2013; Inomata et al., 2014; Zhao et al., 2015,2016). Thus, toluene

and isoprene can provide insight into the roles of water in SOA formation from different kinds of VOCs.

RH has a positive correlation with the mass yield of SOA from aromatics, such as p-xylene (Healy et al., 2009), toluene

(Kamens et al., 2011; White et al., 2014) , o-, p-xylene (Zhou et al., 2011), benzene and ethylebenzene (Jia and Xu, 2014).

This has been mainly attributed to aqueous-phase reactions, such as active uptake of glyoxal in particle water. An exception is

from the study of Cocker et al. ( 2001) who found that the yield of SOA from m-xylene and 1,3,5-trimethylbenzene in the

presence of propene was unaffected by RH (5% and 50%). This is probably due to the presence of propene in their reaction

systems, which can reduce the OH radicals, leading to the decrease in the yield of SOA (Song et al., 2007).

The relationship between RH and SOA formation from isoprene is still not very clear. SOA from isoprene has been widely

studied, as summarized by Carlton et al. (2009). An earlier study from Dommen et al. (2006) showed that RH had little effect

on the SOA yield from isoprene-$NO_x$ ($x$=1, 2) irradiations in the absence of seed particles at 2% and 84% RH. A study from

Zhang et al. (2011a) showed that RH had a negative effect on SOA formation from isoprene-$NO_x$ irradiations with seed

particles of $(NH_4)_2SO_4$ under two RH conditions (15-40% and 40-90%) and ascribed the rise of SOA yield under lower RH to

the enhancement of 2-methylglyceric acid (2-MG) and its corresponding oligomers. Nguyen et al. (2011) found that RH did

not affect the yields of SOA from isoprene in their isoprene-$NO_x$-$H_2O_2$ irradiations without seed particles under dry (2%) and

humid (90%) conditions. Zhang et al. (2012) studied SOA formation from methacrolein (MACR, one of major products from

isoprene) under different ratios of MACR/NO. Their results showed that the effect of RH on formation of SOA depended on

the yields of SOA precursors (e.g. methacryloyl peroxynitrate, MPAN). In addition, isoprene-derived organosulfates (Zhang

et al., 2011a; Zhang et al., 2012) and isoprene epoxydiols (IEPOX) -derived products (Nguyen et al. 2014) are enhanced under

higher RH. A recent study from Lewandowski et al. (2015) showed that the aerosol yield from isoprene-NO irradiations decreased with increasing RH (9% to 49%). The role of water in SOA formation is so complex that more research is still required to understand mechanisms of SOA formation.

Both toluene and isoprene can produce glyoxal during their oxidation processes. Why was the positive effect of RH on the SOA yield of isoprene not observed? We consider that different chemical processes are likely responsible for the different effects of RH on the SOA yields from toluene and isoprene. One of the most important differences between isoprene and toluene reaction systems is oxidation pathways. To clarify the different mechanisms of SOA formed under different humid conditions, this paper presents the experimental results of mass yields and chemical components of SOA from toluene and isoprene under controlled RH conditions, as well as the explanation of the mechanism of SOA formation.

## 2 Experimental section

All the experiments were carried out in a 1.3-m$^3$ FEP reactor (DuPont 500A, USA). The equipment and experimental procedures were similar to our previous works (Jia and Xu, 2014, 2016; Ge et al., 2017). Thus, only a brief introduction is given here. Two ionizing air blowers were around the reactor to remove the electric charge on the surface of the reactor. A light source was provided by black lamps (F40BL, GE, USA), with a center wavelength of 365 nm. The photolysis rate of $NO_2$ was determined to be 0.35 min$^{-1}$ inside the reactor. The humidity was controlled by bubbling the high pure water (18.2 MΩ.cm at 25°C, Millipore /Direct-Q3) with background air. NaCl seed particles were prepared by a constant output atomizer (Model 3076, TSI, USA).

Background air was prepared by a Zero Air Supply (Thermo model 111 and model 1150, USA) with 3 additional VOC traps (BHT-4, Agilent). $NO_x$, $O_3$ and $SO_2$ were measured by corresponding analyzers of Thermo model 42C, model 49C and model 45i (trace level). The concentrations of $NO_x$, $O_3$ and $SO_2$ in background air were determined to be less than 1 ppb. The particles in background air could not be detected with SMPS in the absence of irradiations, but the particle number concentration of $10^4$/cm$^3$ was obtained at an irradiation time of about 5 hrs. It was considered from some experiments and model simulations



that the particles were $H_2SO_4$ (less than 1 μg/m$^3$), which was formed from oxidation of $SO_2$ by OH radicals.

Gas-phase organics were measured with a gas chromatograph-mass spectrometer (GC-MS: Agilent model 7890A GC and Agilent model 5975C mass selective detector, USA), which was equipped with a thermal desorber (Master TD, Dani, Italy). Particle number and mass concentrations were determined by SMPS (TSI model 3936, composed by DMS TSI 3080 and CPC

TSI 3776).

LWC was determined following the method of the reduced Dry-Ambient Aerosol Size Spectrometer (DAASS) (Engelhart et al., 2011). During the dry mode, the SMPS was modified by adding a large diameter Nafion dryer (Permapure MD-700-48F-3, the RH of the sample air can be reduced from 85% to 3.5%) to the sampling inlet and a multi-tube Nafion dryer (Permapure PD-200T-24E-M, the RH of sheath can be reduced from 85% to 7%) to sheath flow. During the humid mode, the humid air in

SMPS was quickly replaced by humid air in the chamber by venting the sheath air at 10 L/min, and then the humid aerosol was measured by SMPS. As a result, the LWC was determined by the difference of the particle mass concentrations between dry and humid modes.

To analyse the chemical components of SOA, the particles ranged from 100 to 650 nm were collected on a 25 mm disk using a Dekati low-pressure impactor (DLPI, Dekati Ltd., Finland) at 10 L/min. Organic functional groups of SOA were measured

by a Fourier transform infrared spectroscopy apparatus (FTIR, Nicolet iS10, Thermo Fisher, USA). The mass spectra of SOA were measured by electrospray ionization high-resolution mass spectrometer (ESI-HRMS, Exactive-Orbitrap mass spectrometer, Thermo Fisher Scientific). The average molecular size information of the humic-like substances (HULIS) present in SOA was determined by UV-Vis spectroscopy (Perkin–Elmer Lambda 25, USA) based on the ratio of E2/E3, in which E2 and E3 denote the absorbance at 250 nm and at 365 nm, respectively (Peuravuori et al., 1997; Duarte et al., 2005).

The liquid reactants of toluene (Xilong Chemical Co., Ltd., 99.8% purity), isoprene (Alfa Aesar, 99.9% purity) or $H_2O_2$ (30 weight % in $H_2O$) were injected into the airline and were evaporated with background air. $NO_2$ (520 ppm in $N_2$, Beijing Huayuan Gas Company) was injected into the reactor directly. For the experiments of the oxidation of isoprene by OH, OH radicals were generated from the photolysis of $H_2O_2$ by UV lights (UVA-340, Q-Lab Corporation, USA). For the experiments of isoprene-$O_3$ dark reaction, $O_3$ was produced by an ozone generator with pure $O_2$ (99.995%). N-hexane (>97% purity, Beijing

Tongguang Fine Chemicals Company) was used as an OH scavenger in the ozonolysis of isoprene. To evaluate the possible contributions of SOA from n-hexane in the ozonolysis of isoprene with n-hexane, two experiments of the irradiations of hexane-$H_2O_2$ were performed for 6 hours, in which no SOA was observed by SMPS under both dry and humid conditions.

The initial conditions and purposes for the experiments are listed in Table 1, most of which are the irradiations of toluene-$NO_2$ and isoprene-$NO_2$. The initial concentrations of isoprene and toluene were about 0.90 and 0.85 ppm respectively, and initial $NO_2$ concentrations were about 320 ppb. At the end of each experiment, isoprene was almost completely consumed after 6-h reactions, and about 400 ppb of toluene was reacted at the end of 7-h reactions. The RH was controlled to be dry (6~10% RH) or humid (78~88% RH) conditions for different experiments. Two sets of experiments with artificially added NaCl seeds (about 10 μg/m$^3$) were performed to quantify the role of particle water on SOA formation in humid toluene and isoprene reactions. To find out how RH affects the oxidation pathways of isoprene by OH and $O_3$ in isoprene-$NO_2$ irradiations, additional experiments of isoprene-$H_2O_2$ irradiations and isoprene-$O_3$ reactions were carried out. The initial $H_2O_2$ and $O_3$ concentrations were around 5 and 1.5 ppm, respectively.

## 3 Results and discussion

### 3.1 RH effects on SOA yields

#### 3.1.1 Determination of LWC

The LWC in particles makes up a great percentage under humid conditions (as shown in Figure 1). To calculate the yield of SOA, the LWC has to be excluded. On the other hand, since LWC was only measured at the end of reaction, to obtain the time evolution of SOA concentrations we needed to deduct LWC during the whole reaction period. The value of LWC depends on chemical components of particles and environmental conditions (temperature and humidity). The volume growth factor (VGF) was used to estimate the contributions of LWC in particles, which was defined by Engelhart et al. (2011) as the ratio of the particle volume at humid air to the particle volume at dry air. Assuming that all the particles are spheres and have the same growth factor, the VGF is equal to the growth factor (GF) cubed as:





$$VGF = \frac{V_{hydrated}}{V_{dried}} = (\frac{D_{hydrated}}{D_{dried}})^3 = GF^3 \tag{1}$$

$V_{hydrated}$ and $V_{dried}$ indicate the total measured volumes of hydrated or dried particles, respectively. $D_{hydrated}$ and $D_{dried}$ are the diameter of hydrated or dried particles, respectively, calculated from their volumes.

*VGF* is determined to be 1.28 (GF=1.09, RH=78%) for the particles from toluene-$NO_2$ irradiations, 1.18 (GF=1.06, RH=79%) from isoprene-$NO_2$ irradiations, 1.40 (GF=1.12, RH=77%) from isoprene-$H_2O_2$ irradiations, and 1.30 (GF=1.09, RH=88%) from isoprene-$O_3$ reaction systems. There have been many studies about the growth factor of SOA. Aklilu and Mozurkewich (2004) gave a GF range of 1.05-1.12 for atmospheric organic particles (79% RH). Stroud et al. (2004) predicted a GF of 1.1 for the organic aerosols from toluene-NO-isopropyl nitrite irradiations at 79% RH. Prenni et al. (2007) reported the GF of 1.065±0.02 at 85% RH for SOA formed from toluene. Jimenez et al. (2009) obtained GF= 1.057±0.02 at 95% RH for SOA from isoprene. In general, our results of GF are in good agreement with previous estimates, indicating that the LWC measured by our modified SMPS is reliable.

### 3.1.2 SOA yields

We assumed that the VGF did not change during the reaction course. Thus, the LWC from toluene and isoprene under humid conditions can be determined by VGF. Figure 2 shows that in touene-$NO_2$ irradiations, the mass concentrations of SOA at 81% RH are much larger than those at 10% RH, with a ratio of maximum mass concentration of SOA at 81% RH to that at 10% being 2.2. However, in isoprene-$NO_2$ irradiations, the mass concentrations of SOA at 80% RH are much lower than those at 7% RH, with the ratio of maximum mass concentration of SOA being 0.57, which is almost the same as that from isoprene-$O_3$ reactions. For isoprene-$H_2O_2$ irradiations, the mass concentrations of SOA from humid conditions are generally larger than those under dry conditions. Nevertheless, the maximum mass concentration of SOA from humid conditions is 177.9 μg/m$^3$, which is close to that from dry conditions.

The mass yield of SOA generally increases with time. The maximum yields during the experimental course were used for the following discussion. The mean maximum yields of SOA from toluene were obtained to be 5.58±0.76% (dry) and 8.97±0.84% (humid) respectively (Figure 3). Our results are within the range obtained by other investigators (Kamens et al., 2011; Odum



et al., 1997; Ng et al., 2007). Previous studies (Kamens et al., 2011; Zhou et al., 2011; Jia & Xu 2014; Wang et al., 2016) mainly ascribed the positive effect of RH on SOA yields from aromatics to LWC, which can enhance the formation of SOA by aqueous reactions, such as reactive uptake of glyoxal in aerosol water. Our yields of SOA from toluene are smaller than those from Ng et al. (2007) (around 11% at 4% RH) and Hildebrandt et al. (2009) (11-17% at 21% RH), which is probably

due to the additional and excessive OH radical sources (HONO or $H_2O_2$) used in their experiments.

A negative effect of RH on SOA yields was observed in the systems of isoprene-$NO_2$-hv and isoprene-$O_3$. The mean maximum yields from isoprene-$NO_2$ irradiations are reduced from 3.14±0.35% (dry) to 1.19±0.38% (humid) (Figure 3). This negative RH effect is in good agreement with the corresponding results from Zhang et al. (2011) and Lewandowski et al. (2015). The yields of SOA from our isoprene-$O_3$ reactions are 3.00% (dry) and 1.40% (humid), which are quite close to the results from

the isoprene-$NO_2$ system. This shows that the ozonolysis of isoprene is a key pathway influencing the SOA formation in isoprene-$NO_2$ irradiations. RH has little effect on the SOA yields from isoprene-$H_2O_2$ irradiations. The maximum yields of SOA were determined to be 7.7% (dry) and 7.8% (humid) from photooxidation of isoprene-$H_2O_2$, which are in good agreement with the results (around 8%) of isoprene-$H_2O_2$ irradiations under dry conditions from Clark et al. (2016). A similar yields (7%) of SOA from photooxidation of isoprene-$NO_x$-$H_2O_2$ was also obtained in the results from Nguyen et al. (2011). Based on the

experimental conditions in Nguyen et al. (2011), we estimated that for their reaction system over 99% of isoprene was oxidized by OH and the remaining 1% by $O_3$ by using simulations based on the Master Chemical Mechanism (MCM v3.3.1, website: http:// mcm.leeds.ac.uk/MCM, Jenkin et al., 2015). Thus, the reaction system by Nguyen et al (2011) can be considered to be closer to isoprene-$H_2O_2$ system. These results show that high RH can reduce the maximum yields of SOA from the reaction channel of isoprene with $O_3$ ($O_3$ channel) and that RH has little effect on the maximum yields from the reaction channel of

isoprene with OH (OH channel).

In our isoprene-$NO_2$ irradiations, based on the MCM simulation, the amount of isoprene oxidized by OH, $O_3$ and $NO_3$ is 59%, 25% and 16% at the end of reactions, respectively. For these three oxidation channels, RH has little effect on SOA yields from OH channel oxidization. $NO_3$ is similar to OH, and only $O_3$ channel is greatly influenced by RH. The maximum possibility is that $O_3$ channel can produce SCIs that can be consumed by water. Thus, although most of isoprene was oxidized by OH and



the SOA yield from the OH channel was over 2 (5) times greater than that from the $O_3$ channel under dry (humid) conditions, the $O_3$ channel was still a major pathway influenced by water vapor in the isoprene-$NO_2$ system, which will be discussed in the following section.

### 3.2 UV-Vis spectra of SOA

Toluene and isoprene have a similar yield of glyoxal and a close value of *VGF* in this study. Thus, it would be reasonable to expect that RH should have a positive influence on the SOA yield from isoprene. However, the yield is greatly reduced in the humid systems of isoprene-$O_3$ and isoprene-$NO_2$, and almost unchanged in isoprene-$H_2O_2$ irradiations. Zhang et al. (2011a) and Nguyen et al. (2011) both reported that oligomers were greatly reduced under humid conditions, and considered that high RH suppressed the oligomerization reactions with water as a product. The molecular sizes of SOA can reveal the degree of oligomerization reactions. Thus, we used the UV-Vis spectra to determine the molecular size information of SOA indirectly. Larger molecules displayed higher absorbance in longer wavelength regions as surmised by Mostafa et al. (2014) All the spectra are characterised with a continuous absorption that increases with decreasing wavelength from 200 nm up to about 1100 nm (Figure 4), which indicates the presence of conjugated double bond molecules (such as oligomers).

We used the UV-Vis spectra to determine the molecular size information of SOA, which can provide information about oligomerization degree of molecular through the ratio of E2/E3 indirectly. Lower E2/E3 ratios are associated with higher molecular weight (Peuravuori et al., 1997; Duarte et al., 2005). The ratios of E2/E3 are 1.27 (1.53), 1.29 (2.25), 1.69 (1.97) and 1.55 (1.73) under dry (humid) conditions from our systems of toluene-$NO_2$, isoprene-$NO_2$, isoprene-$O_3$ and isoprene-$H_2O_2$, respectively. The E2/E3 ratios show that high RH can indeed reduce molecular sizes of SOA and suppress the oligomerization reactions generating water as a product. Nevertheless, if the suppression of the oligomerization reactions under humid conditions is the main reason for the decrease in SOA yield from isoprene, why is the maximum yield from isoprene-$H_2O_2$ irradiations unchanged under humid conditions. In addition, oligomers have been also identified as important products of SOA from aromatics, and water is a byproduct during oligomerization process (Lim et al., 2010; Gaston et al., 2014; Kalberer et al., 2004). However, a negative effect of RH on SOA yield from aromatics has never been observed. Thus, there must be an





intrinsic mechanism regarding the influences of RH on the SOA yield from isoprene.

### 3.3 IR spectra of SOA

### 3.3.1 Toluene

Figure 5 (a and b) shows the typical infrared spectra of SOA from the irradiations of toluene under both dry and humid

conditions. The prominent features on SOA spectra are the board hydrogen bonded O−H stretching, the carbonyl C═O stretching, the organic nitrate (ONO$_2$) bands, the C−OH bands of alcohols or polyalcohol. The bands all greatly increase by over 2 times as RH increases from 10% to 81%. These changes of band strength with RH are quite similar to the changes of the SOA mass yield with RH.

To further reveal the chemical properties of the increased products formed from humid conditions, the SOA sample from

humid conditions was evaporated. First, we found that the IR spectrum of SOA almost did not change after being heated at 100°C for 15 minutes. Then the sample was further evaporated at 110°C for 15 minutes. After the evaporation at 110°C, the major absorption bands were considerably changed. As a result, the spectrum is almost the same as the spectrum of SOA collected under dry conditions (Figure 5c). The major reduced absorptions are from the O−H and C−OH bands (Figure 5d). These bands are assigned to hydrates of glyoxal and other water soluble compounds in SOA (Volkamer et al, 2009; Lim et al,

2010; Kamens et al., 2011; Jia and Xu, 2014; Wang et al., 2016). Therefore, it is considered that alcohols (such as hydrates) are major contributors to toluene SOA under humid conditions.

LWC is an important factor that can greatly influence the contribution of SOA from aqueous reactions. The maximum LWC was measured to be about 44 μg/m$^3$ from humid toleuene-NO$_2$ irradiations. To determine the role of LWC in SOA formation, extra LWC was introduced into the reaction system by adding 10 μg/m$^3$ of NaCl particles. The initial LWC was determined to

be 30 μg/m$^3$, and maximum LWC was 74 μg/m$^3$ during 6 hours of reaction. The major changes for SOA were all the bands assigned to O−H, C═O and C−OH (Figure 5e), which were enhanced by 50%, 29% and 35% respectively, and the mass concentrations of SOA were increased by 16%, compared to those in humid conditions without extra LWC. This demonstrates that the increase of LWC can greatly enhance the formation of SOA from hydration of glyoxal. Therefore, it is concluded from



our study that the formation of SOA from toluene is controlled by LWC under humid conditions, and that most of SOA is

formed by aqueous reactions in touene-$NO_2$ irradiations.

### 3.3.2 Isoprene

#### 3.3.2.1 Isoprene-$NO_2$ system

The spectra of SOA from the irradiations of isoprene-$NO_2$ are characterized by the high abundance of $C{=}O$ and $ONO_2$ groups

(Figure 6). There are 3 bands assigned to different kinds of $C{-}O$ or $C{-}O{-}O$ groups in the region of 927 -1243 $cm^{-1}$ under dry

conditions (Pretsch et al., 2009). These bands are indicators of alcohols and polymeric structures (Czoschke et al., 2003). Thus,

oligomers and organic nitrates are dominant species in SOA. Under humid conditions, the absorption intensities of the bands

($O{-}H$, $C{=}O$, $ONO_2$, $C{-}O$ or $C{-}O{-}O$) are all reduced by 2 times. The tert-nitrate can hydrolyze in particle water by the

replacement of tert$-ONO_2$ to $-O-H$ group (Liu et al., 2012). Because such replacement hardly changes the vapor pressure of

corresponding species (Pankow et al., 2008), newly formed alcohols should remain in the aerosol phase. We also did extra

experiments to test the hydrolysis of organic nitrates. After the SOA sample from dry isoprene-$NO_2$ irradiation was exposed

to humid air (90% RH) for 1 hour, we did not find any apparent change in $ONO_2$ group. Meanwhile, the peak height ratios of

$ONO_2/O{-}H$ from SOA are almost the same under dry and humid conditions. Thus, the hydrolysis of nitrates is not the major

reason for the decrease of particle phase organic nitrates. It also indicates that aerosol phase oligomers can hardly be influenced

by RH. Then, high RH likely inhibited the formation of particle-phase organics by reducing the oligomerizations in the gas

phase (e.g. SCI-derived oligomers).

RH generally enhances SOA formation by the aqueous reactions. Similarly, the aqueous reactions also exist in isoprene-$NO_2$

irradiations. However, the maximum LWC from humid isoprene-$NO_2$ irradiations was measured to be 8 μg/m³ at the end of

reaction, which is much smaller as compared to 44 μg/m³ in toluene irradiations. Taking glyoxal as an example, although the

maximum concentrations of glyoxal were simulated to be 39 ppb in isoprene-$NO_2$ irradiations, which is only 60% of its

maximum concentrations of 65 ppb from toluene irradiations, due to the limitation of LWC, the SOA from the aqueous

reactions was significantly reduced in isoprene-$NO_2$ irradiations. To further confirm the role of LWC, we did an additional

experiment with NaCl seeds (initial LWC of 30 μg/m³) in isoprene-$NO_2$ irradiations. The results show that the absorptions of

the bands from O−H and C−OH increase by 20% to 30% as compared to that without additional LWC (Figure 6). It is true

that increasing LWC can indeed enhance SOA formation in isoprene-$NO_2$ irradiations; however, the absorptions of C=O,

$ONO_2$ and C−O from dry conditions are still 2 times larger than those from the experiment with extra LWC. This demonstrates

that the increase in SOA through aqueous reactions is far less than the decrease due to $H_2O$-related reactions under humid

conditions. Thus, high water vapor can probably inhibit some key processes responsible for SOA formation from isoprene-

$NO_2$ irradiations, which will be discussed in the following contents.

**3.3.2.2 Isoprene-$H_2O_2$ and Isoprene-$O_3$ systems**

To determine which process responses to the decrease of SOA under humid conditions from isoprene-$NO_2$ irradiations, IR

spectra of SOA from the OH and $O_3$ channels were studied respectively. Since O−H-containing products from terpene are

more enriched from the OH channel oxidation than from the $O_3$ one (Calogirou et al., 1999), here the absorption ratio of O−H

to C=O was used to examine the difference between the $O_3$ and OH oxidation channels. Higher values of O−H/C=O ratio

should be expected to be more from the OH channel than from the $O_3$ channel.

The IR spectra of SOA from the isoprene-$H_2O_2$ system are characterized by strong absorptions of hydrogen bonded O−H and

C−OH and weak absorption of C=O under both dry and humid conditions (Figure 7 top), with the peak height ratios of O−H/C

=O being 1.63 (dry) and 1.45 (humid), which strongly supports that alcohols or polyalcohols are major components of SOA

from isoprene-$H_2O_2$ irradiations. Under humid conditions, the peak at 1090 cm$^{-1}$ assigned to C−O−C group from esters is

slightly decreased, while the band at around 3200 cm$^{-1}$ from O−H absorption is broadened as compared to the dry condition.

It indicates that esters (e.g. oligomers) decrease while the compounds containing O−H increase under humid conditions.

Nevertheless, the relative abundances of O−H, C=O and C−OH groups are almost the same between dry and humid conditions,

which shows a weak effect of RH on SOA from isoprene-$H_2O_2$ irradiations as compared to isoprene-$NO_2$ irradiations. In OH

channel, isoprene can be oxidized to form $RO_2$ ($ISOPO_2$). If there is no NO, $ISOPO_2$ will be further oxidized to isoprene

epoxydiols (IEPOX) by OH and $HO_2$ radicals. IEPOX are key intermediates of SOA in isoprene-OH reactions (Surratt et al.,





2010). Under dry conditions, IEPOX can be adsorbed on $H_2SO_4$ seeds to form polyalcohols (e.g. 2-methyltetrols) through acid-catalyzed heterogeneous reactions, which can further form oligomers by esterification (Lin et al., 2012). Under humid conditions, IEPOX can be absorbed into particle water to produce polyalcohols (Nguyen et al., 2014). In addition, the decrease of C−O−C group indicates that the formation of oligomers is inhibited by the abundance of particle water as discussed in

section 3.2.2, which is in agreement with the result of Lin et al. (2014). Because polyalcohols (dominant) and IEPOX-derived oligomers are all in the aerosol phase, the total mass concentration of SOA does not change much under humid conditions in isoprene-$H_2O_2$ irradiations. In other words, RH does not change the partition of IEPOX in our experimental conditions. This is consistent with the result of Riva et al. (2016) that water has a weaker impact on IEPOX-derived SOA yield.

As reported by Gaston et al. (2014), the reaction probability ($\gamma_{IEPOX}$) of IEPOX into acidic particles decreased with the increase

of RH due to the dilution effect. Thus, the yield of SOA is expected to show a negative correlation with RH. However, both the yield and IR spectra of SOA formed from OH pathway in isoprene-$H_2O_2$ irradiations are weakly dependent on RH under our experimental conditions. $H_2SO_4$ particles are formed in background air in our experiments. The mass of $H_2SO_4$ particles is less than 1 $\mu g/m^3$. When liquid water content increases from 1 $\mu g/m^3$ to the maximum 54 $\mu g/m^3$ under humid conditions, the pH value is estimated to be in the range of 2 to 3.7. In this pH range the value of $\gamma_{IEPOX}$ is still large enough (Gaston et al.

2014), so that partition is not the limited step for IEPOX uptake. Similarly, SOA formation can also be enhanced by acid-catalyzed reactions in toluene-$NO_2$ irradiations (Jang et al., 2002). However, our experimental results show that the yield of SOA is increased rather than decreased at higher RH. Therefore, the acidity dilution effect by higher RH is not remarkable under our experimental conditions.

In the isoprene-$O_3$ systems, if the bands from $ONO_2$ are excluded, both the shape and band intensities of IR spectra of SOA

are quite similar to those of SOA from isoprene-$NO_2$ irradiations. All the bands assigned to O−H, C═O and C−O are reduced by over 2 times under humid conditions (bottom panel of Figure 7). The ratios of O−H/C═O are 0.36 (0.44) under dry (humid) conditions. The results are consistent with our expectation that lower ratios of O−H/C═O should be in SOA from the ozonolysis of isoprene. Since OH radicals were well removed in our experiments, SCI became the key intermediates of SOA. The C−O−O group is an indicator for the participation of SCI in SOA from the ozonolysis of isoprene. The C−O−O group is





very apparent under dry conditions, which decreases by 60% under humid condition. Oligomer products in SOA have been

suggested to be formed by the reactions of n (n=1-10) SCI with $RO_2$ in the ozonolysis of small enol ethers and trans-3-hexene

(Sadezky et al., 2008; Zhao et al., 2015). Thus, SCI derived oligomers are also deduced to be the key components in SOA

from the $O_3$ channel of isoprene. To examine the influence of RH on oligomer formation from SCI, the reactions of SCI with

$RO_2$ and with their products (n=1-10) were added to MCM. The rate constant for these reactions was set to be $5 \times 10^{-12}$ cm$^3$

molecule$^{-1}$ s$^{-1}$ (Vereecken et al., 2012). When RH increases from 10% to 88%, the consumption of SCI by water increases

from 13% to 58%, while the SCI-derived oligomers decrease from 87% to 42%. The reaction products of SCI with $H_2O$ have

relative high vapor pressures as compared to oligomers, so they are mainly in the gas phase. Therefore, humid condition can

reduce the SOA formed by SCI-related reactions in the isoprene-$O_3$ systems.

In isoprene-$NO_2$ irradiations, the ratios of O−H/C＝O are 0.35 (0.36) under dry (humid) conditions, which are almost the same

as the corresponding values in isoprene-$O_3$ but totally different from the values in isoprene-$H_2O_2$. The yields, IR spectra (ratios

of O−H/C＝O) and the influence of RH on SOA production from the isoprene-$O_3$ system are almost the same as those from

isoprene-$NO_2$ irradiations. In isoprene-$NO_2$ irradiations, even though 60% of isoprene was oxidized by OH, because of the

presence of NO, most of $ISOPO_2$ from oxidation of OH could be quickly consumed by NO to form MPAN (around 15 ppb

under both dry and humid conditions) and other products, leading to the decrease of IEPOX from 224.0 ppb (in isoprene-$H_2O_2$)

to 41.2 ppb (in isoprene-$NO_2$). MPAN is one of key precursors of SOA from isoprene under high $NO_x$ conditions (Surratt et

al., 2010). With additional OH sources, MPAN can be oxidized by OH to form SOA products (such as 2-methyltetrols) (Surratt

et al., 2010; Lin et al., 2013; Nguyen et al., 2015). Both the results from Nguyen et al. (2014) and MCM-simulation further

show that if there are enough OH radicals, most of MPAN can be further oxidized by OH to produce SOA precursors of

epoxides (e.g., hydroxymethyl-methyl-a-lactone (HMML), methacrylic acid epoxide (MAE)), such as in the Nguyen et al.

(2011) work with $H_2O_2$ as an extra OH source. The yield of MACR is generally greater in isoprene-$NO_2$ irradiations and

isoprene-$O_3$ systems than that in isoprene-$H_2O_2$ irradiations. MACR can react to form MPAN in the presence of $NO_2$, which

can be oxidized by OH to yield epoxides and then 2-MG and related oligomers. 2-MG-derived oligomers can be enhanced

under lower RH (Zhang et al., 2011). However, since there were no extra OH sources in our systems, MCM simulations show

that only 12% (24%) of MPAN was oxidized by OH to produce HMML and MAE. It means that only 3~4 ppb of MPAN could

form SOA, which is too small to explain the yields of SOA in isoprene-$NO_2$ systems. If we simply assume that the

concentrations of SOA were proportional to the IEPOX concentration as in isoprene-$H_2O_2$ irradiations, over 70% of SOA

should come from IEPOX in dry or humid isoprene-$NO_2$ irradiations. However, the IR spectra of SOA from dry or humid

isoprene-$NO_2$ are totally different from those in isoprene-$H_2O_2$ irradiations. On the contrary, they are similar to those from

isoprene-$O_3$ system. Thus, IEPOX is not the major contributor to SOA in isoprene-$NO_2$ systems. This clearly demonstrates

that MPAN and IEPOX related reactions were not dominant pathways for SOA formation in our isoprene-$NO_2$ irradiations.

On the other hand, similar to the isoprene-$O_3$ system, SCI-related reactions in the isoprene-$NO_2$ system were probably key

pathways. To quantify the formation and the RH effect of SOA in isoprene-$NO_2$ irradiations, the SCI-related reactions were

added into MCM. Simulations show that when RH increases from 7% to 80%, the concentration of SCI-derived oligomers

decreases by 44% because of the reaction of SCI with water. Therefore, without extra OH sources, the formation of SOA is

still controlled by the SCI.

**3.4 Mass spectra of isoprene SOA**

To further determine whether SCI-derived oligomers are the major components of SOA from isoprene-$NO_2$ irradiations, the

high resolution mass spectra of SOA under dry and humid conditions were obtained with ESI-HRMS (Figure 8). The mean

molecular size of SOA was reduced from 352 under dry conditions to 295 under humid conditions, which is in good agreement

with the results by UV/Vis spectra. The peaks on the spectrum show highly regular mass differences, especially in the range

of 300 ~800 m/z, which is a typical structure for polymers or oligomers (Kalberer et al., 2004). The total intensity of peaks in

the range of 300 to 800 m/z under dry conditions is reduced by 75% as compared to that under humid conditions. This

demonstrates that oligomers are probably a major component of SOA from isoprene-$NO_2$ irradiations, which are greatly

reduced under humid conditions. The mass spectrum of SOA from the ozonolysis of isoprene is similar to the one from

theisoprene-$NO_2$ system. The spectrum of SOA from isoprene-$H_2O_2$ (Figure 9) shows a very different feature from that SOA

from isoprene-$NO_2$. It does not reveal obviously regular structures of the peaks for oligomers.




### 3.4.1 Analysis of oligomers with KMD method

To further characterize whether SCIs are the major building blocks of the oligomers in SOA from isoprene-NO$_2$ irradiations,

a Kendrick mass defect (KMD) analysis was used. The KMD analysis is a standard method to visualize the complex organic

mass spectra (Kendrick, 1963). The Kendrick mass (KM) is converted from the IUPAC mass M by multiplying a factor

of $NM_{base}/M_{base}$ (i.e., the factor is 14.00000/14.01565 for the base unit of CH$_2$) using equation (2). $NM_{base}$ is the exact mass

$M_{base}$ rounded to the nearest integer. KMD is calculated as the difference between the nominal KM (NKM) and KM using

equation (3). The basic principle of KMD method is that a homologous series of compounds differing only by a number of

base units have identical KMD values. Thus, the KMD analysis allows for the rapid identification oligomers by a plot of KMD

*versus* KM, in which homologous compounds can line up in the horizontal direction. Since the KMD analysis has a great

advantage to clearly determine the molecular composition of hundreds of individual compounds in SOA samples, it has been

applied extensively for complex SOA sample analyses using HR-MS (Reinhardt et al., 2007; Walser et al., 2008; Nguyen et

al. 2010, 2011; Nizkorodov et al., 2011). In addition, since different series of homologous oligomers may have similar KMD

values, the KMD data need to be pre-sorted by z* value that is calculated by equation (4) (Hsu et al., 1992).

$$KM = M \times \frac{NM_{base}}{M_{base}} \qquad (2)$$

$$KMD = NKM - KM \qquad (3)$$

$$z^* = \text{modulo}\left(\frac{NM}{NM_{base}}\right) - NM_{base} \qquad (4)$$

### 3.4.1.1 Base units of oligomers: SCIs

There are 16 kinds of SCIs produced in isoprene-NO$_2$ irradiations based on MCM v3.3.1 simulation, in which CH$_2$OO (CH$_2$O$_2$,

with the yield of 50.1%), MACROO (C$_4$H$_6$O$_2$, 18.3%), MVKOO (C$_4$H$_6$O$_2$, 12.2%), MGLOO (C$_3$H$_4$O$_3$, 11.3%) and GLYOO

(C$_2$H$_2$O$_3$, 2.6%) account for 95% of total SCIs. To explain that these SCIs exist in SOA as base units of CH$_2$O$_2$, C$_4$H$_6$O$_2$,

C$_3$H$_4$O$_3$ and C$_2$H$_2$O$_3$, a wide set of other base units (OH, CO, NO$_2$, ONO$_2$, CH$_2$, CH$_2$O and COO) are also included for KMD

analysis. The ratio of oligomers with a given base unit to total mass is defined to characterize the contribution of different base



units to SOA. It should be pointed out that large uncertainties exist in the estimate of relative contributions of different units because of the poor quantification performance using ESI-MS techniques. Due to the cross containing of units in oligomer molecules, the sum of ratios is larger than 100%. The compounds with a same base unit (M-[base unit]$_n$, $n$=0,1,2,3…) that contains at least 3 compounds ( the maximum n bigger than 2) are considered as one class of oligomers. The results show that

only the ratios of oligomers with the base units of $CH_2O_2$, $C_4H_6O_2$, $C_3H_4O_3$ and $C_2H_2O_3$ are proportional to the yields of corresponding SCIs from the isoprene-$NO_2$ system. Figure 10 displays the correlation diagram between the ratios of oligomers (with $CH_2O_2$, $C_4H_6O_2$, $C_3H_4O_3$ and $C_2H_2O_3$ as repeating units) to total mass and top 5 SCI yields ($CH_2OO$, MACROO,MVKOO, MGLOO and GLYOO). It shows that the ratios linearly increase with increasing yields under both dry and humid conditions. Thus, this demonstrates that the oligomers with $CH_2O_2$, $C_4H_6O_2$, $C_3H_4O_3$ and $C_2H_2O_3$ repeat units are from contribution of

these SCIs of $CH_2OO$, MACROO (& MVKOO), MGLOO and GLYOO in the isoprene-$NO_2$ system. Therefore, these 5 SCIs are chosen as the base units for KMD analysis, which shows that the ratios of compounds containing SCI units are reduced by 45% on average as RH increases from 7% to 85%. This is also in good agreement with MCM simulation of decrease in SCI-derived oligomers by 44% and with the decrease of intensity of peroxide C−O−O absorption in FTIR (Figure 6). In addition, the KMD analysis is also used to determine the chemicals of oligomers in SOA from isoprene-$H_2O_2$. All the above base units

are tested, and the results show that $CH_2O$-containing oligomers are the major products in SOA, and the lengths of oligomers are much shorter than those from isoprene-$NO_2$. The maximum repeat unit number of n is less than 3 in most families of oligomers in SOA from isoprene-$H_2O_2$. By contrast, the maximum value of n is larger than 5 in oligomer families of SOA from isoprene-$NO_2$. This indicates that SCIs incline to produce long chain oligomers.

### 3.4.1.2 Base unit of $CH_2OO$

It has been considered that the $CH_2OO$ radical can serve as an oligomer unit in SOA from the ozonolysis of ethylene (Sakamoto et al. 2013). $CH_2OO$ has the highest yields (50.1%) of all the SCIs from isoprene. The KMD analysis shows that the mass of $CH_2OO$-containing oligomers account for 46.2% (29.4%) of total mass on the MS under dry (humid) conditions. Figure 11 displays the selected mass spectra of oligomers with $CH_2OO$ as chain units and their corresponding KMD plots under dry and

humid conditions, which shows that both the length of oligomer chains and the number of oligomers are greatly reduced under humid conditions. The number of oligomers under humid condition is reduced by 64% as compared to dry conditions. Another feature of $CH_2OO$-based oligomers is that the molecular sizes of their monomers are larger than 300 (C14-C17), which probably come from other oligomers formed during reactions. This indicates that most $CH_2OO$-based oligomers are formed in the particle phase.

### 3.4.1.3 Base units of MACROO & MVKOO

Figure 12 shows the mass spectra and KMD plot of oligomers with $C_4H_6O_2$ (MACROO and MVKOO) as base units. The KMD analysis results show that the ratio of $C_4H_6O_2$- based oligomers to total compounds is 39.7% (17.2%) under dry (humid) conditions. In addition to $CH_2OO$, MACROO and MVKOO based oligomers have the second highest contribution to SOA among all the SCIs. Similar to $CH_2OO$, the maximum number of chain units is 6 in oligomers from $C_4H_6O_2$. However, the molecular size of monomers is much smaller than that from $CH_2OO$. The most frequent monomer is $C_3H_6O_2$ as shown in Figure 12. Based on Chemspider database and MCM simulation, we deduced that $C_3H_6O_2$ is from hydroxyacetone (ACETOL), which is the most abundant carbonyl-containing products in isoprene-$NO_2$ reaction system. The maximum concentration of ACETOL is over 90 ppb based on our experimental conditions. $C_2H_2O_3$ is deduced to be glyoxylic acid that is one of products from isoprene irradiation. SCIs can react with carbonyl and alcohol products (e.g., ACETOL), $RO_2$ and $H_2O$. However, different from the isoprene-$O_3$ system, MCM simulations show that most of $RO_2$ is consumed by NO in isoprene-$NO_2$ irradiations. Thus carbonyl and alcohol products become the major monomers for SCI oligomerizations.

An addition of a C−O−O group can change the vapor pressure of oligomers (containing $n$ SCI units) by a factor of $2.5 \times 10^{-3}$ (Pankow et al., 2008). The vapor pressures of SCI-derived oligomers (e.g. $C_3H_3O_2$-$[C_4H_6OO]_n$) are estimated to be less than $10^{-7}$ atm ($n \geq 2$) and $10^{-12}$ atm ($n \geq 4$). The compounds can self-nucleate as their vapor pressures are less than $10^{-9}$ atm (Kamens et al., 1999). It indicates that the initial particles in the $O_3$ oxidation channel of isoprene are formed by the self-nucleation of oligomers ($n \geq 4$). The oligomers with $n \geq 2$ probably further condensed on these particles. Thus, MACROO or MVKOO based oligomers can be formed in the gas-phase (e.g. reaction 4 and 5). With the increase of chain units, these oligomers can either





self-nucleate or further oligomerise in the aerosol phase.

glyoxylic acid ($C_2H_2O_3$)    SCI (MACROO, $C_4H_6O_2$)    oligomers    (4)

hydroxyacetone ($C_3H_6O_2$)    SCI (MACROO, $C_4H_6O_2$)    oligomers    (5)

### 3.4.1.4 Other base units of oligomers

5  It is worth noticing that the yield of $CH_3OC_3H_3OO$ ($C_4H_6O_3$) is only 1.4% based on the MCM simulation. However, the

contribution of $C_4H_6O_3$ to oligomers is as high as 25.3% (9.0%) under dry (humid) conditions (Table 2). Thus, $C_4H_6O_3$ is not

totally from $CH_3OC_3H_3OO$. 2-MG usually serves as molecular tracers for isoprene SOA (Kleindienst et al., 2007). As reported

by Zhang et al. (2011) and Nguyen et al. (2011), $C_4H_6O_3$ was the repeated unit of 2-MG's corresponding oligomers in SOA

from isoprene-$NO_x$ irradiations. $C_4H_6O_3$ is formed from dehydration of 2-MG in oligomers. Thus, most of $C_4H_6O_3$ base

10  oligomers can be contributed by 2-MG in our work. The ratios of $CH_2OO$, MACROO and MVKOO based oligomers are

almost 2 times larger than that from 2-MG under both dry and humid conditions. Thus, even though the ratio of $C_4H_6O_3$ base

oligomers was decreased by 65% as RH increased from 7% to 85%, 2-MG derived oligomers would not be the major reason

for the decrease of SOA yield from isoprene-$NO_2$ irradiations. In addition to $C_4H_6O_3$, $CH_2$ and $CH_2O$ based oligomeric

compounds also have high ratios in isoprene-$NO_2$ systems, which have been also reported as the most prominent units in SOA

15  products from the ozonolysis of isoprene in the Nguyen et al. (2010) study under dry conditions. However, different from SCI

based oligomers, the ratios of $CH_2$ and $CH_2O$ based oligomers decreased by 18% and 14% as RH increased from 7% to 85%,

respectively. Thus, the reduction of SCI based oligomers is the major reason for the decrease of SOA yields from isoprene-

$NO_2$ photooxidations.



### 3.5 Mechanisms for the different roles of water in isoprene-NO₂ systems

### 3.5.1 Vapor wall loss vs SCI-H₂O reaction

It is noted that the wall loss of semi-volatile organic compounds (SVOC) can lead to the underestimation of the yield of SOA (Matsunaga & Ziemann 2010; Loza et al., 2010; Zhang et al., 2014; Yeh & Ziemann 2015; Ye et al., 2016; Palm et al., 2016; Krechmer et al., 2016; La et al., 2016; Nah et al., 2017). Since SCI-derived oligomers are the major products of SOA from isoprene-NO₂ irradiations, a question arises about which process is dominant for the reduction of SOA production under humid condition, wall loss of SCI related oligomers ( in gas-phase) or the reaction between SCI and $H_2O$. The MCM simulation shows that SCIs are so reactive that most of them are consumed by reactions before they are lost to the wall (Figure 13). The percentage of the SCI consumed by $H_2O$ was increased from 6% to 46% as RH increased from 5% to 85% due to the extremely high concentration of gas $H_2O$. The removal of SCI by $H_2O$ (85%RH) can lead to a decrease of SCI –derived oligomers by 43% as compared to 5%RH. The result is comparable with the decrease of SOA yields of 62%) from isoprene-NO₂ irradiations. Meanwhile, as discussed in the previous section, the vapor pressure of SCI-derived oligomers were very low that they were ready to condense on particles. The upper limit of wall loss rate constant of $4.8 \times 10^{-4}$ s⁻¹ for SVOC was calculated from the equation given by McMurry and Grosjean (1985), while the condensation rate constant of SVOC to the particles was calculated to be over 0.65 s⁻¹ in our study based on the equation from La et al (2016). This indicates that the condensation rate of gas-phase oligomers to particles is much fast than that to the wall. Therefore, the reactions between SCIs and $H_2O$ rather than the wall loss of SVOC is the major cause for the decrease of SOA formation from isoprene-NO₂ irradiations in this work.

### 3.5.2 Effects of water on SOA formation: O₃ vs OH

Our results clearly show that the different effects of RH on SOA yields originate from the oxidation channels (Figure 14). Both the OH and O₃ channels can well explain the differences of results in isoprene-NO₂ irradiations from the Zhang et al. (2011) and Nguyen et al. (2011) studies. In the Zhang et al. study, there were no additional OH radical sources in their systems. Thus, the SOA was mainly from the O₃ channel. Similar to our isoprene-O₃ systems, a negative effect of humidity on SOA yield was observed in their work. In the Nguyen et al. work, due to sufficient OH radical source, over 99% of isoprene was oxidized by

OH, and SCI concentrations were very low. Even though high $NO_x$ was used, most of MPAN could be further oxidized by OH to produce epoxides. Therefore, SOA was mainly from the OH channel in Nguyen et al.'s work. This is why the yield of SOA in their work was not influenced by RH. Our results obviously show that SOA is formed by reactive uptake of SOA precursors (e.g. IEPOX) in the OH channel, and by the condensation of SCI-derived oligomers in the $O_3$ channel. In the presence of $NO_2$,

the formation of SOA is also controlled by the SCI-related reactions without extra OH sources. However, the SCI-related reactions (SCI-derived oligomers) can be inhibited by high water vapor. In previous studies, SOA was usually modeled based on the vapor pressures of SVOC, which only considers the effect of temperature. Our study strongly suggests that RH is also a key factor in SOA formation.

## 4 Conclusion

Opposite effects of RH on SOA formation from the irradiations of toluene and isoprene have been elucidated in our work. Different influences of RH on both SOA yields and mean molecule size demonstrate the different mechanisms related to SOA formation from the irradiations of toluene and isoprene. High RH can greatly enhance the SOA formation in toluene-$NO_2$ system, so that the maximum yields of SOA from toluene increased from 5.58% (dry) to 8.97% (humid). FTIR spectra show that the increased part of SOA under humid conditions was mainly contributed by aqueous reactions of water soluble products

(e.g. glyoxal). Different from toluene-$NO_2$ irradiations, water has a complex role in isoprene systems. In isoprene-$H_2O_2$ irradiations systems, RH has no remarkable effects on SOA yields. FTIR spectra show that water can inhibit the oligomerization reactions; however, polyalcohols were still the major products in both dry and humid conditions from isoprene-$H_2O_2$ irradiation, which was mainly from the reactive uptake of IEPOX in the presence of $H_2SO_4$ particles from background gas. In isoprene-$O_3$ and isoprene-$NO_2$ irradiation systems, high RH has a negative effect on SOA yields, which

decreased from 3.14% (dry) to 1.19% (humid). According to the FTIR, ESI-HRMS, KMD analysis and MCM simulations, it is considered that the oligomers with SCIs as base units were the major products of SOA in isoprene-$O_3$ systems and isoprene-$NO_2$ irradiation systems. Under humid conditions, the SCIs can be consumed by water in the gas-phase, leading to the decrease





of the formation of oligomers from SCIs.

**Acknowledgments**

This work was supported by the National Natural Science Foundation of China (No. 41375129) and National Key R&D Program of China (2017YFC0210005)

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

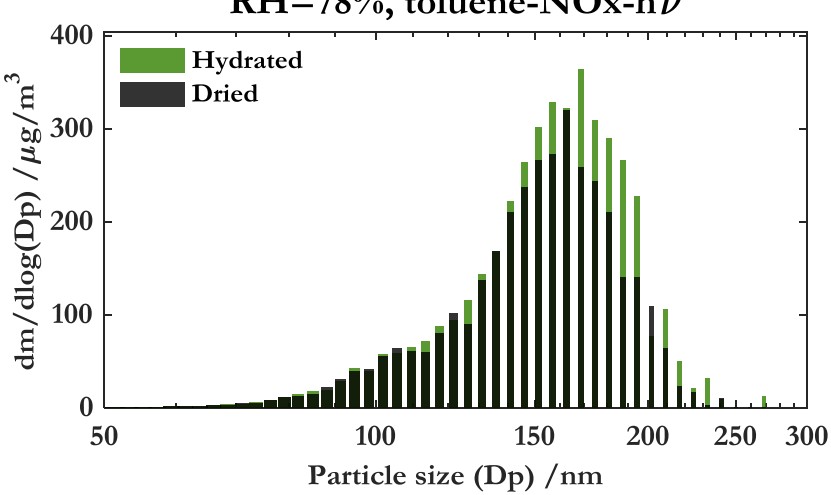

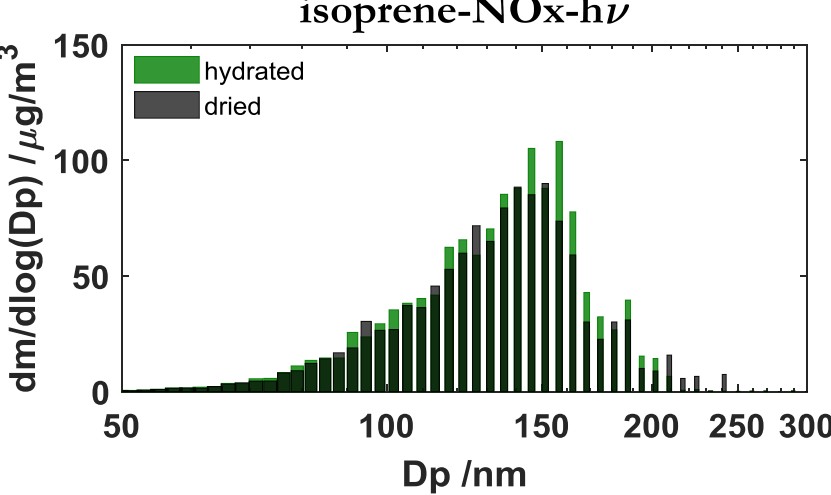





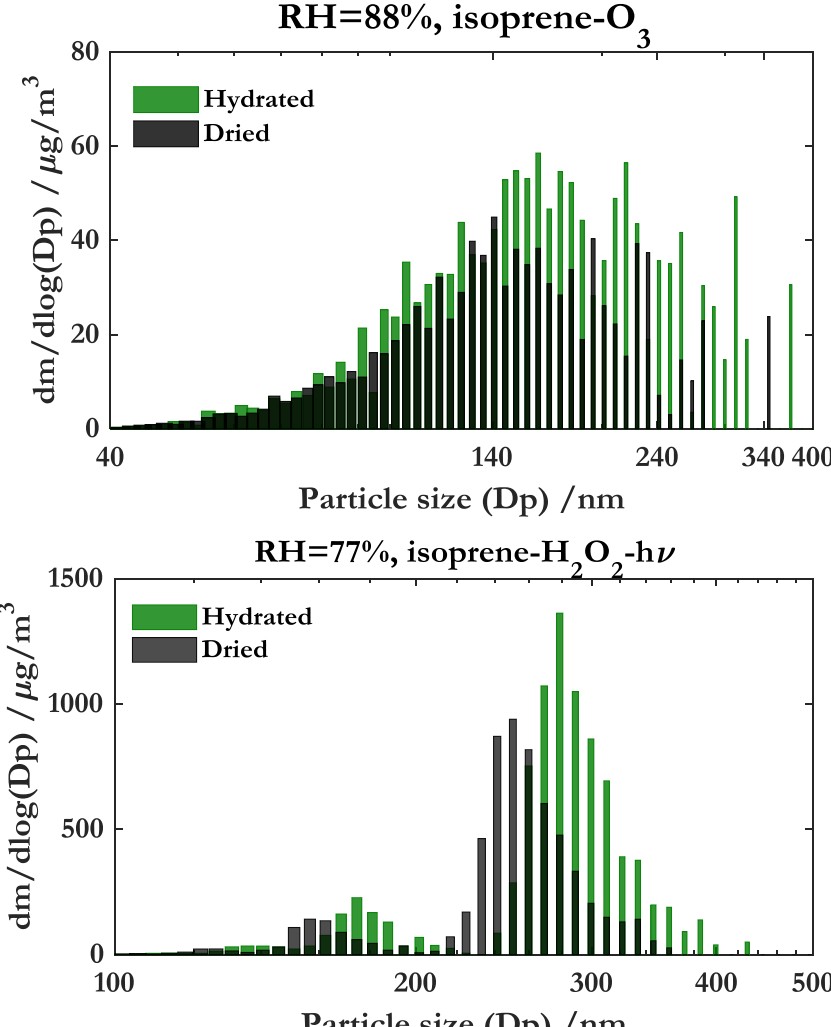

**Figure 1: Mass cocnentration ditributions of both dryed and hydrated particles from both toluene and isoprene systems at 3 hr after the initiation of reaction.**





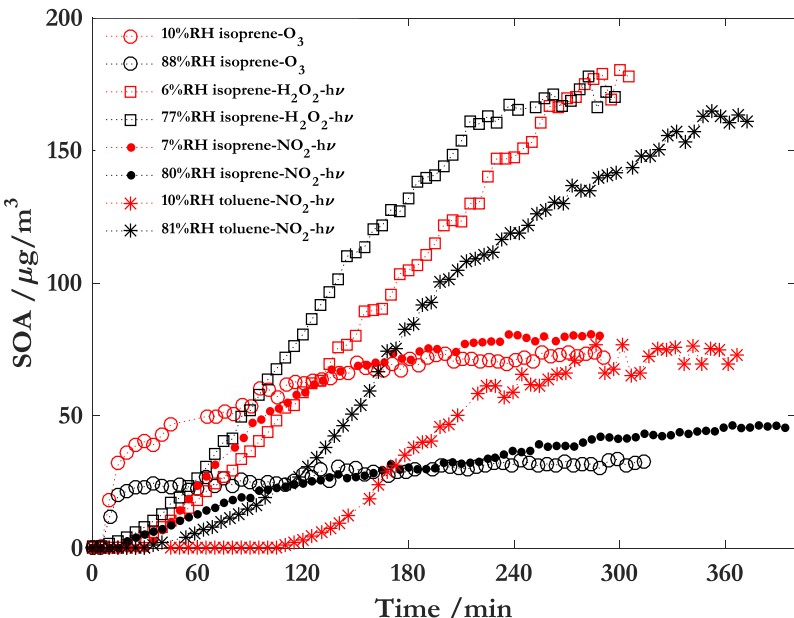

**Figure 2: Mass concentration time profiles of SOA from different toluene and isoprene reaction systems under dry and humid conditions. An SOA density of 1.4 g/cm$^3$ was used and applied to the SMPS mass correction (Dommen et al., 2006; Sato et al., 2007). The wall loss rate constant of particles was less dependent on RH conditions, so an average value $4.8 \times 10^{-3}$ /min was used to correct the SOA formation.**

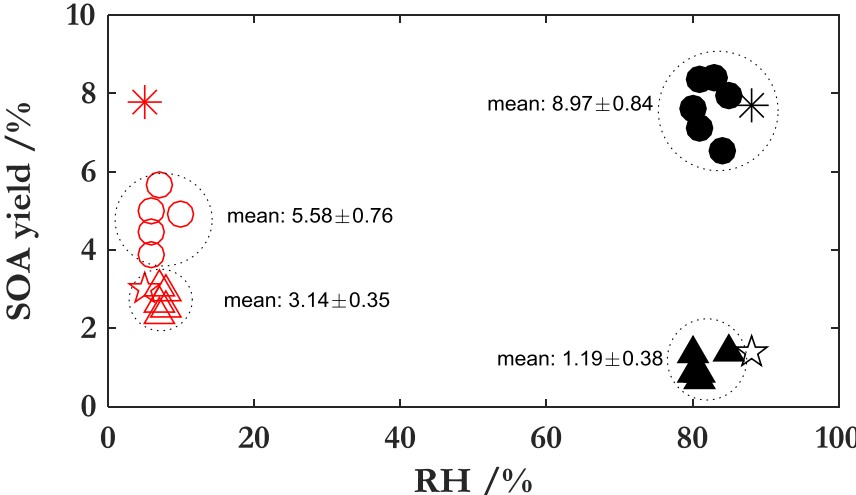

**Figure 3: Maximum yields of SOA from toluene and isoprene under dry (red color) and humid (black color) conditions (○: toluene-NO$_2$-hv; △: isoprene-NO$_2$-hv; ☆: isoprene-O$_3$; ✳: isoprene-H$_2$O$_2$-hv).**



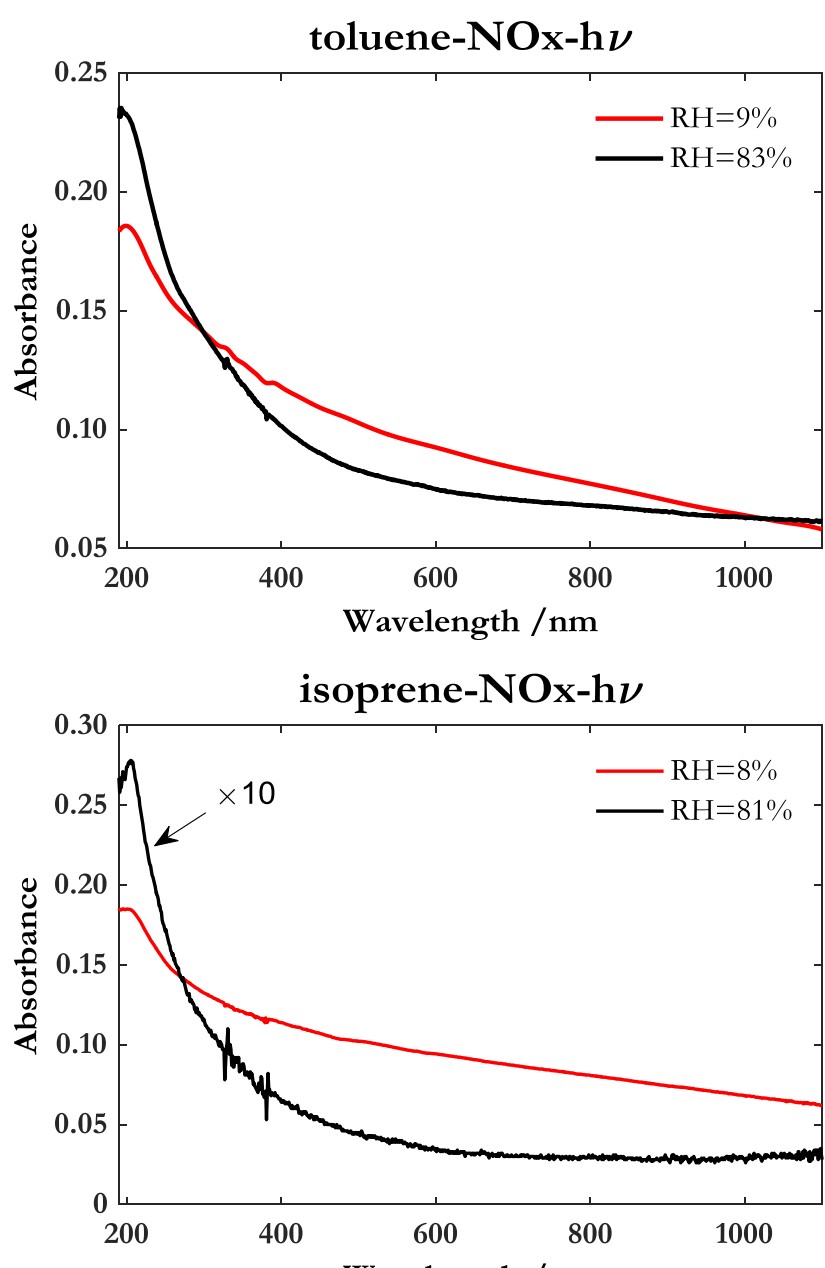

Figure 4: UV-Vis spectra of SOA from toluene and isoprene photooxidations under dry and humid condition.



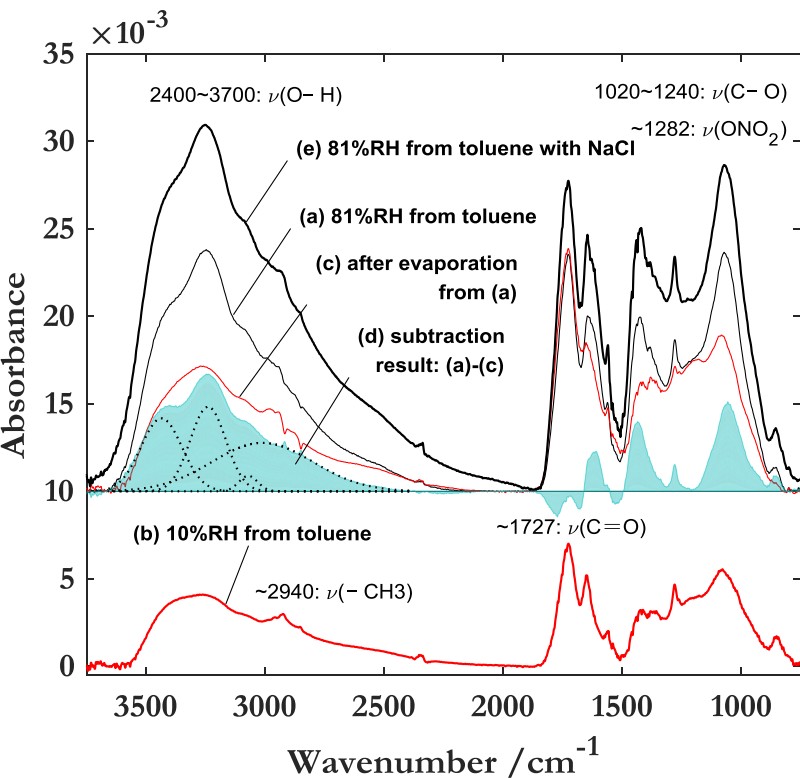

**Figure 5: The infrared spectra of SOA from toluene irradiations under humid (top: a) and dry (bottom: b) conditions; the infrared spectrum of (c) was obtained after evaporation of SOA from (a) for 15 min at 110 °C; the difference spectrum between (a) and (c) is in blue area (d); the infrared spectrum of (e) is with extra LWC by NaCl seeds. Main bands are hydrogen bonded O−H stretch in alcohols or acids (2400-3700 cm⁻¹), the carbonyl (C=O) band at 1727 cm⁻¹, the organic nitrous (ONO₂) bands at 1636, 1278, and 855 cm⁻¹, the C−OH bend band at 1423 cm⁻¹, and the C−OH stretch at 1080 cm⁻¹.**



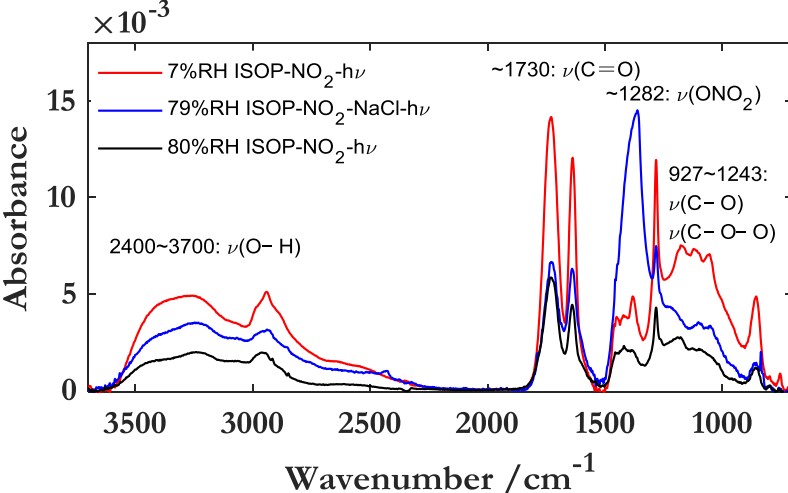

**Figure 6: Infrared spectra of SOA from isoprene irradiations under dry and humid conditions. The bands at 1636, 1282, and 855 cm⁻¹ are from ONO₂. The bands at 1170 and 1121 cm⁻¹ are assigned to C−O−C in oligomers or C−O in carboxylic acids, and the band at 1055 cm⁻¹ is from alcohols. The absorption shoulder from 927 to 1080 cm⁻¹ is assigned to C−O and O−O in peroxide group (C−O−O) (Pretsch et al., 2009).**



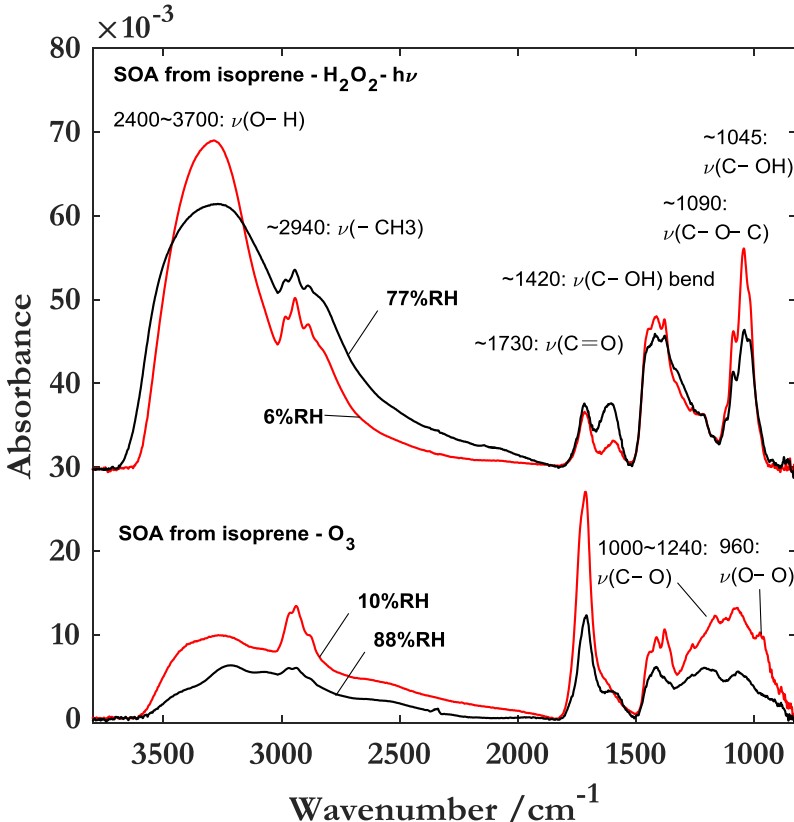

**Figure 7: Infrared spectra of SOA from isoprene with different oxidants under dry and humid conditions. The bands at 1051 and 960 cm⁻¹ are assigned to C−O and O−O groups in peroxide C−O−O (Pretsch et al., 2009).**



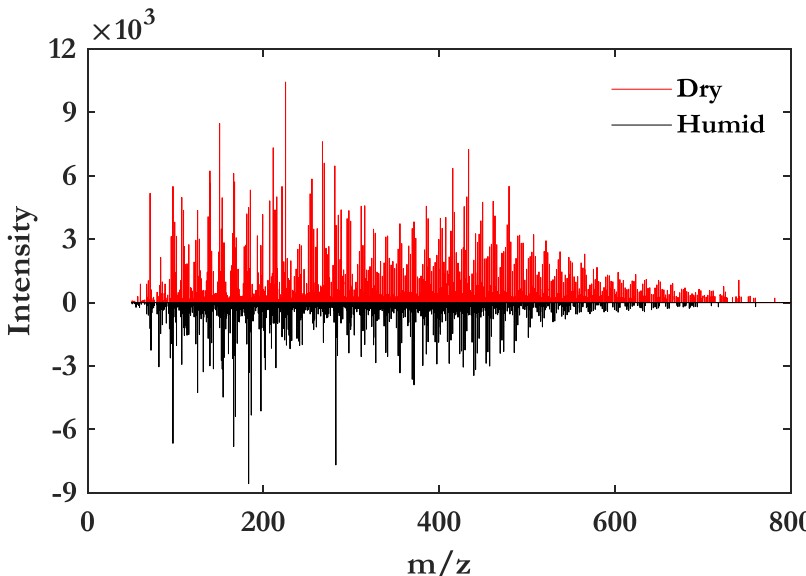

**Figure 8: Positive ion mode ESI-Orbitrap mass spectra of SOA from isoprene-NO₂ irradiations under dry (top panel) and humid (bottom panel) conditions.**

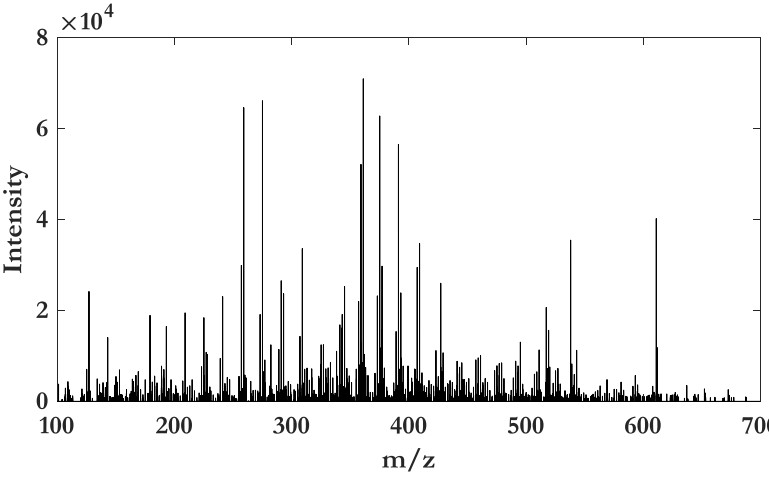

**Figure 9: Positive ion mode ESI-Orbitrap mass spectra of SOA from isoprene-H₂O₂ irradiations under dry conditions.**

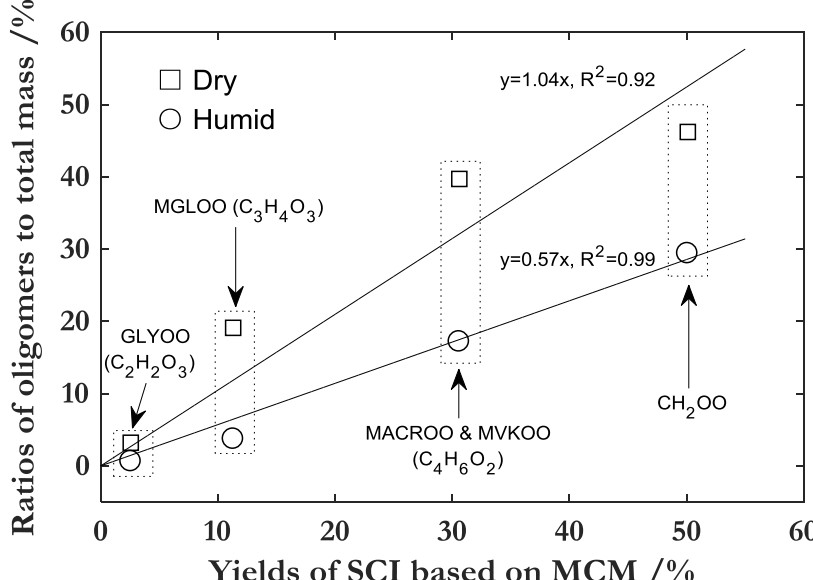

Figure 10: The correlation of yields of top 5 SCIs (CH2OO, MACROO & MVKOO, MGLOO and GLYOO) and the ratios of CH2O2, C4H6O2, C3H4O3 and C2H2O3 based oligomers to total mass under dry and humid conditions from isoprene-NO2 irradiations. Only the oligomers that belong to families (M-[SCI]n, n=0,1,2,3…) with maximum number of n larger than 2 was taken into consideration.





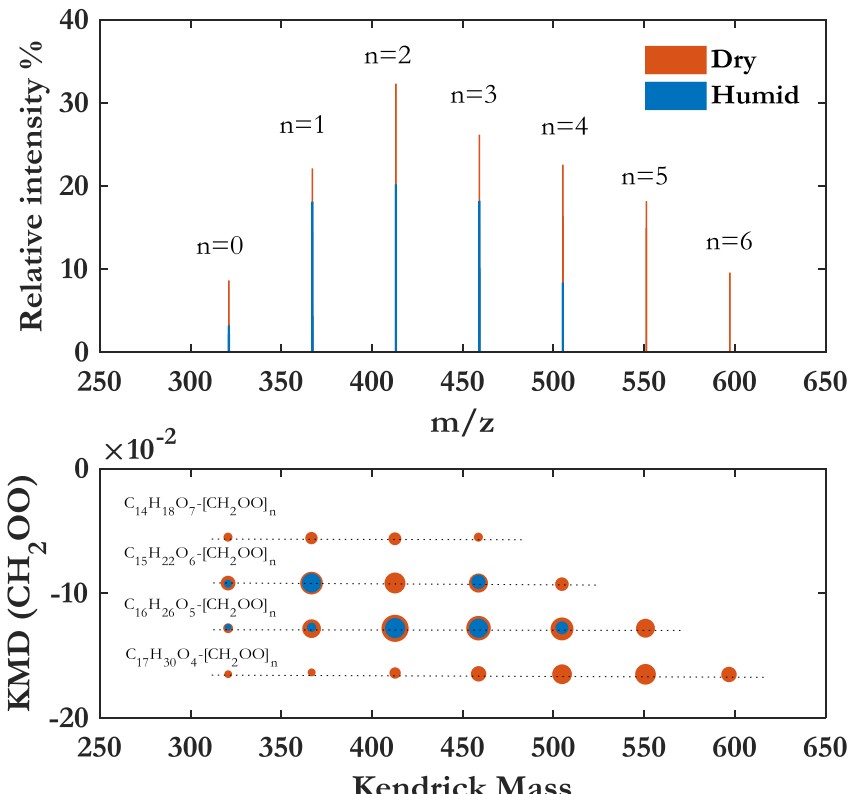

**Figure 11: Positive mode mass spectra of oligomers with CH₂OO as chain units in SOA from isoprene-NO₂ irritations under dry and humid conditions (*top*) and corresponding plots of KMD(CH₂OO) versus nominal KM(CH₂OO) (*bottom*). Horizontal lines connect the family of compounds with an equal elemental composition differing only by a [CH₂OO]ₙ groups (n=0,1,2,3…).**



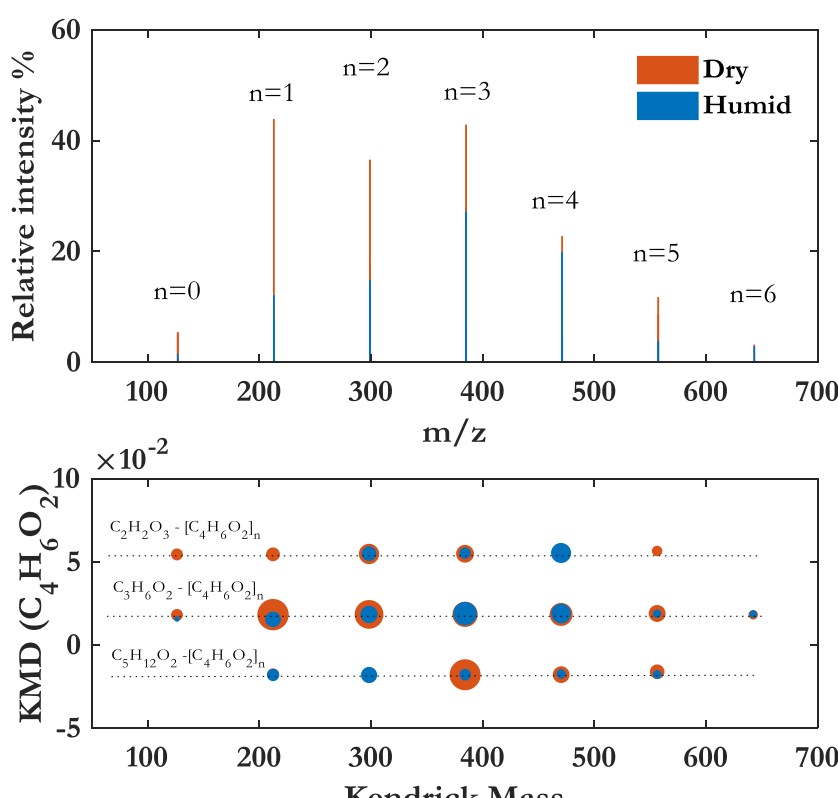

**Figure 12: The mass spectra of oligomer with C$_4$H$_6$O$_2$ (MACROO and MVKOO) as repeating unit and their Kendrick plots using C$_4$H$_6$O$_2$ as Kendrick base. Species separated by C$_4$H$_6$O$_2$ groups fall on horizontal lines.**



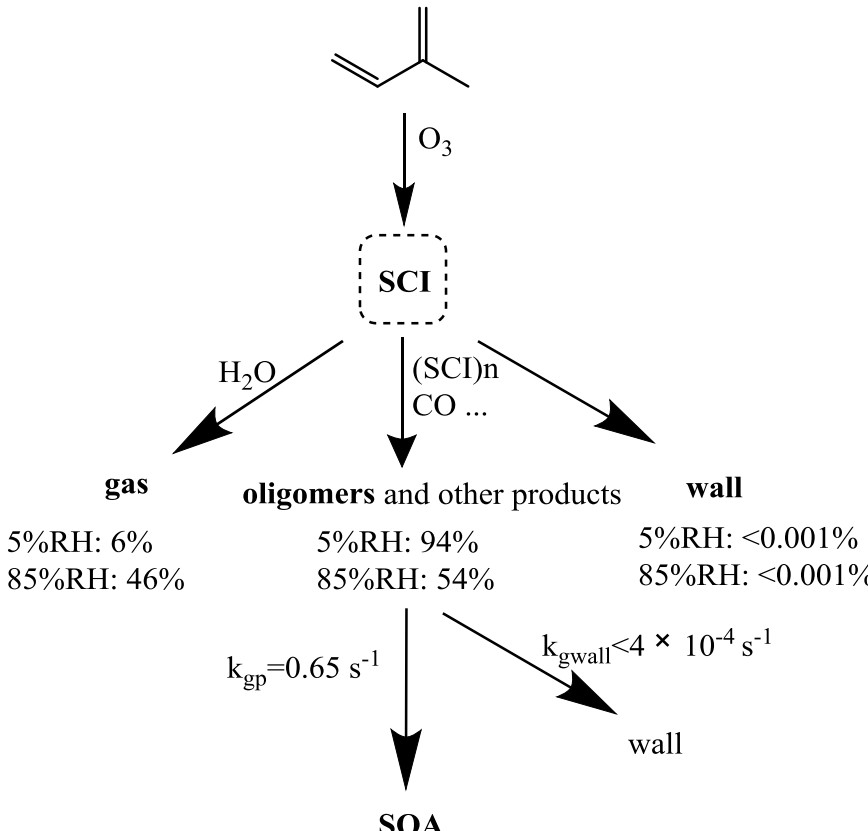

**Figure 13: Wall losses vs gas-phase reaction between H₂O and SCI**


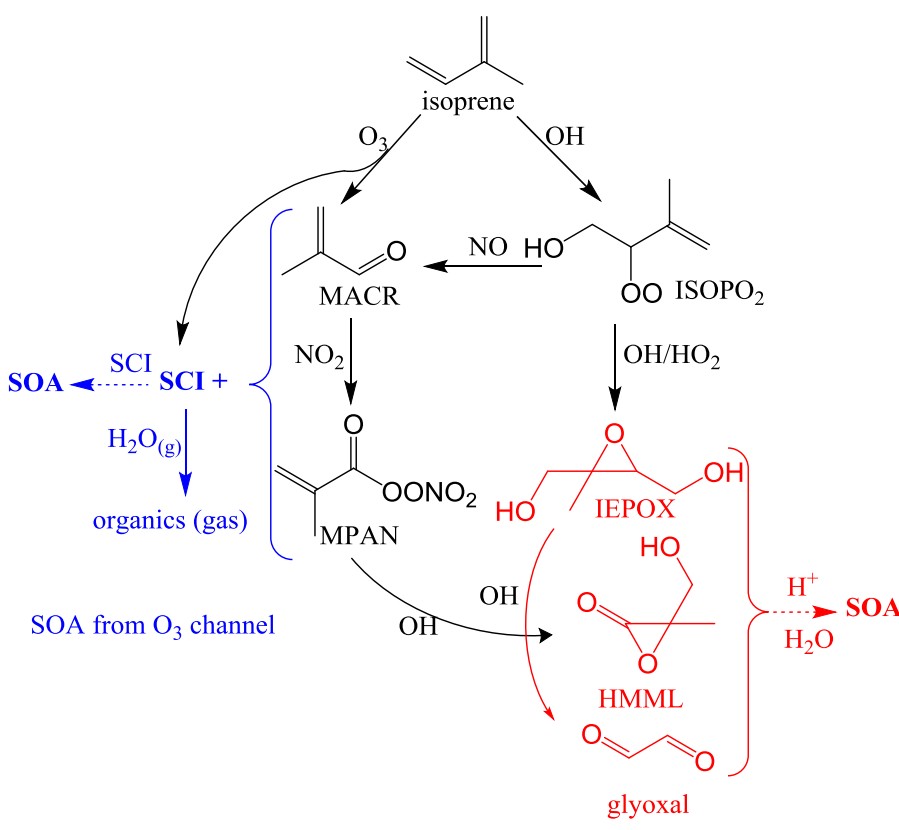

**Figure 14: Mechanisms for SOA formation from the O₃ and OH oxidation channels of isoprene.**

**Table 1 Expermental conditions of toluene and isoprene irradiations**

| VOCs | # | T /K | RH /% | VOC /ppm | NO /ppb | NO₂ /ppb | Aim |
|------|---|------|-------|----------|---------|----------|-----|
| Toluene | 1 | 304 | 6 | 0.915 | 4.3 | 307.7 | SOA size & yield |
| | 2 | 304 | 85 | 0.804 | 5.3 | 303.6 | |
| | 3 | 304 | 84 | 0.933 | 1.5 | 293.9 | |
| | 4 | 304 | 6 | 1.037 | 0.2 | 323.3 | |
| | 5 | 302 | 6 | 0.879 | 12.0 | 328.0 | |
| | 6 | 303 | 10 | 0.917 | 1.9 | 326.7 | FTIR |
| | 7 | 303 | 81 | 0.846 | 7.8 | 301.0 | |
| | 8 | 304 | 7 | 0.930 | 9.0 | 325.0 | FTIR with NaCl seeds |
| | 9 | 303 | 81 | 0.906 | 10.0 | 334.0 | |
| | 10 | 304 | 9 | 0.914 | 10.6 | 386.5 | UV/Vis |


|          | 11 | 304 | 80 | 0.910 | 7.7  | 364.1 |              |
|----------|----|-----|----|-------|------|-------|--------------|
|          | 12 | 305 | 79 | 0.927 | 12.1 | 288.1 | LWC by FTIR  |
|          | 13 | 305 | 79 | 0.918 | 10.6 | 294.7 | LWC by SMPS  |
|          | 14 | 302 | 7  | 0.896 | 7.0  | 353.0 |              |
|          | 15 | 302 | 85 | 0.804 | 6.0  | 364.0 | size and     |
|          | 16 | 301 | 7  | 0.850 | 0.0  | 311.7 | yield        |
|          | 17 | 302 | 80 | 0.844 | 0.3  | 308.5 |              |
|          | 18 | 303 | 7  | 0.901 | 0.0  | 299.5 |              |
|          | 19 | 303 | 81 | 0.799 | 0.3  | 270.2 | FTIR         |
|          | 20 | 302 | 80 | 0.828 | 0.2  | 273.0 |              |
| Isoprene | 21 | 301 | 8  | 0.790 | 0.0  | 283.1 |              |
|          | 22 | 303 | 9  | 0.873 | 3.0  | 301.0 | FTIR with    |
|          | 23 | 303 | 79 | 0.827 | 4.0  | 325.0 | NaCl seeds   |
|          | 24 | 303 | 8  | 0.823 | 0.1  | 332.5 | UV/Vis       |
|          | 25 | 303 | 81 | 0.877 | 0.3  | 363.0 |              |
|          | 26 | 305 | 81 | 0.831 | 1.5  | 288.8 | LWC by FTIR  |
|          | 27 | 304 | 78 | 0.823 | 0.5  | 313.5 | LWC by SMPS  |
|          | 28 | 303 | 7  | 0.810 | 0.2  | 295.1 | ESI-HRMS     |
|          | 29 | 303 | 85 | 0.804 | 0.5  | 290.2 | ESI-HRMS     |

**Table 2 List of major base units and their corresponding ratios of oligomers to total mass from SOA in isoprene-NO₂ irradiations.**

| Base unit | Unit name | SCI yields by MCM | Ratio of oligomers to total mass | | |
|-----------|-----------|-------------------|--------|--------|--------|
|           |           |                   | Dry /% | Humid /% | Delta /% |
| $CH_2O_2$ | $CH_2OO$ | 50.1 | 46.2 | 29.4 | 36.3 |
| $C_4H_6O_2$ | MACROO & MVKOO | 30.6 | 39.7 | 17.2 | 56.7 |
| $C_3H_4O_3$ | MGLOO | 11.3 | 19.1 | 3.7 | 80.4 |
| $C_2H_2O_3$ | GLYOO | 2.6 | 3.2 | 0.6 | 80.1 |
| $C_4H_6O_3$ | dehydrated 2-MG (or $CH_3OC_3H_3OO$) | (1.4) | 25.3 | 9.0 | 64.6 |
| $CH_2$ | - | - | 76.4 | 62.7 | 17.9 |
| $CH_2O$ | - | - | 78.5 | 67.6 | 13.8 |

