# Peer review of "Different roles of water in secondary organic aerosol formation from toluene and isoprene"

_Atmospheric Chemistry and Physics, 2017_

## Referee Comment (RC1) · Anonymous Referee #1 · 6 Jan 2018

This manuscript on different roles of water in toluene and isoprene SOA formation presents interesting results. Overall, the results are consistent with previous studies: toluene SOA formation is enhanced by higher RH due to aqueous-phase reactions and isoprene SOA formation is enhanced by lower RH due to oligomer formation. The main finding suggests that the isoprene SOA (ozonolysis and NO2 irradiation) formation is largely controlled by the "SCI → oligomer" process that is influenced by water. The manuscript is well written and provides sufficient measurements supporting the main conclusion, but a few issues need to be addressed before considered publication in ACP.

Main Comments:
1. On page 2, line 14, the authors claimed that "the relationship between RH and SOA formation from isoprene is still not very clear". However, from the previous studies listed in this paragraph, it appears that a pretty clear understanding has been established: (1) oligomers have always been found to enhance under low RH; and (2) SOA yield may not always enhance accordingly, because the competing pathways such as glyoxal SOA or organosulfate formation are increased under high RH and the overall SOA mass vary with the relative contributions from different pathways. Thus, it is problematic to say the RH effect on isoprene SOA is "not very clear". Instead, what remains a question is, whether the known MAE-derived oligomers explain all oligomers. The authors' findings fit right into this question and should thus set up this direction in the introduction. Also, in the introduction, the authors should provide more details of previous work that has found SCI as key oligomeric chain unit and give more credit to these studies (e.g., Sadezky et al., 2008 ACP and a few other studies). Right now, it was only briefly mentioned that SCI-derived products have been found in SOA, but considering the results of this work, the relationship between SCI and oligomers should be elaborated in more detail.

2. In section 3.1, the authors stated that the similar RH effect on isoprene SOA from ozonolysis and NO2 irradiation experiments suggest that the ozonolysis of isoprene is a key pathway influencing the SOA formation in isoprene-NO2 irradiations. This is not necessarily true. In the isoprene-NOx experimental conditions used in Zhang et al. (2011 ACP), most isoprene were reacted with OH before ozone was formed. In the Lewandowski et al. (2015 ACP) study, continuous mode was used and thus minimal ozone is expected to form. Nguygen et al. (2011 ACP) does not produce much O3 and still observe enhanced oligomers under dry conditions (despite SOA yield was not affected). In these studies, ozonolysis play a small role in isoprene oxidation and observed similar results (or the authors may provide MCM simulations and prove otherwise). Even for the studied condition, as the authors stated, isoprene reacts with OH (59%) and O3 (25%) and the SOA yield from the OH channel was over 2 (5) times greater than that from the O3 channel under dry (humid) conditions. Thus, at most 10-15% of isoprene SOA are from the O3 channel. This contradicts to the observed doubled SOA formation under dry conditions if all the RH effect is from the O3 channel.

3. It appears that the authors used MCM simulation a few times in this study. I feel it might be worthwhile adding a section describing what they did. Particularly, SCI + RO2 reactions and SCI-derived oligomer formation were added by the authors in the mechanism. Quantitatively, how much could the modeled total SCI-derived oligomers account for the measured total isoprene

SOA? Other references include, for example, the authors claiming MPAN+OH needs extra OH source. This is not the case in the Lin et al. (2013 PNAS). Without clearly show the MCM simulations, it is hard to draw the conclusion. These are very important aspects in the results and should be provided in more detail.

4. The mass spectral data suggest "the SCI based oligomers are almost 2 times larger than that from 2-MG". However, the data are not quantitatively calibrated. It is unclear which is more important.

Overall, the authors reporting isoprene SCI-derived oligomers in SOA is well justified and important. These oligomers can be important fractions of oligomers under isoprene high-NOx conditions and might explain previously observed oligomers in isoprene SOA besides those derived from MAE. However, based on the results provided in this work, it is not convincing that the SCI-derived oligomers are the dominant part in isoprene SOA, especially under isoprene-NO2 irradiation conditions.

Minor comment:
1. Page 2, line 16. The term "isoprene-NOx (x=1, 2)" is odd. Suggest using "isoprene-NOx (NO and NO2)".

2. Page 2, line 19. Nguyen et al. (2011) did not find a RH influence on SOA yield, but did observe enhancement of oligomers under low RH, which is consistent with Zhang et al. This should be mentioned in this sentence.

3. Figures. The 4 plots in Figure 1 need to be numbered. The authors have consistently used red colors for low-RH results and black for high-RH results in many figures which is very helpful. I suggest use the same color scheme in Figures 6 (replace the blue line with a black dashed line), 10 (use colors for symbols and lines), 11, and 12.

4. Page 7, line 3. There are other possible reasons accounting for the lower SOA yield. For example, this study uses higher temperature than the two previous studies mentioned.

5. Page 8, line 5. Is this based on MCM simulation? Please add clarification in the text.

6. Page 13, line 17. "MPAN can be oxidized by OH to form 2-methyltetrols" is wrong. The products are 2-methylglyceric acid and related oligomers. This is repeated by later sentences in this paragraph. The authors should consider consolidate/rewrite this paragraph.

7. Page 14, line 6. "Thus, IEPOX is not the major contributor to SOA in isoprene-NO2 system. This clearly demonstrates that MPAN and IEPOX related reactions were not dominant pathways for SOA formation in our isoprene-NO2 irradiations". The logic in this sentence is problematic. It cannot suggest MPAN reactions are not important in isoprene-NO2 irradiations. Again, a figure showing simulated MPAN, SCI, and a few other important products is needed at least.

8. Page 18, line 9. $C_4H_6O_3$ is not formed from dehydration of 2-MG, but MAE (Lin et al., 2013 PNAS).

---

## Referee Comment (RC2) · Anonymous Referee #2 · 9 Feb 2018

This article examines the influence of relative humidity (RH) on selected toluene and isoprene SOA formation pathways. Through yield calculations, the author discovers toluene-NO2 system has a higher SOA yield at high humidity environment while isoprene has a lower yield at higher RH for NO2 and O3 pathways. This study involves various techniques and methods such as Master Chemical Mechanism (MCM) modeling, Fourier Transform Infrared Spectroscopy (FTIR), ultraviolet-visible spectroscopy, and EST-HRMS. The author concludes that the higher yield of toluene SOA at high RH is due to aqueous reactions of water-soluble components, while the decreasing yield of isoprene SOA under NO2 and O3 pathways is due to reduced oligomer formation from Criegee intermediate (SCI) reacting with water.

This study combines both experimental techniques and modeling to support the above

arguments that the authors are trying to make. The modeling results add more credibility to the arguments. The manuscript is comprehensive and provides useful information about the chemical composition of SOA formed from various pathways. However, there are a few points that the author should consider when making these arguments, and these issues should be addressed before the article is published in ACP.

Major Comments

First, the author draws the conclusion that RH does not have an effect on isoprene SOA yield under the OH oxidation pathway. However, Gaston et al. (2014) clearly show that RH has an effect on isoprene epoxydiols (IEPOX) reactive uptake into the acidic sulfate particles due to the dilution effect at higher RH. Because IEPOX is one of the main oxidation products of isoprene under OH pathway, the yield should show a negative correlation with RH. But the author's experimental results seem to not agree with those of past studies. I believe the main reason for non-RH dependence observed in this study is that the acidity of the H2SO4 particles generated by this study is too low, resulting in RH not having an additional acidity dilution effect. The author should at least mention the discrepancy between this work and other studies, and point out limitations of this study. For instance, page 7, line 19-20: the author made a conclusion that RH had little effect on the yield of isoprene with OH reactions. Please note that this is probably true for generating self-nucleated isoprene SOA, but may not be true for isoprene SOA that are formed on acidic sulfate seed particles, as shown by a few other studies (Gaston, Riedel et al. 2014, Riedel, Lin et al. 2015, Zhang, Chen et al. 2018). These studies suggest that RH has an important effect on the formation of isoprene-OH-heterogeneous generated SOA, due to the change of RH affecting the acidity of the seed particles. The author should include the literature and clarify the difference of RH effects on experiments because the seed particles in this study probably do not have enough concentration to undergo full heterogeneous reactions.

Page 7, lines 10-11: the author concludes that ozonolysis is the key pathway influencing SOA formation in isoprene-NO2 systems. How does the author draw such

conclusion? It seems that the author just compares the yields of isoprene-NO2 with isoprene-O3 under dry and humid conditions, and finds out the yields were similar. But similar yields do not mean that the reaction mechanisms are similar. The ozone concentrations in the two conditions could be different. There might also be synergistic or competing mechanisms for SOA formation when NO2 and O3 are both present. Besides, the author also shows on page 7, lines 21-22 that NO2 and O3 pathway each contributes to isoprene oxidation for certain extent. Therefore, the author cannot rule out the possibility that NO2 and O3 pathways having similar yields, rather than O3 pathway dominates the NO2 pathway. I think the author should either provide more evidence in this part to show isoprene-O3 really dominates, or make a more conservative the conclusion and state the limitations.

Page 7, line 23: the author points out that RH has little effects on NO3 pathway. Please list any evidence (such as references) to support this point.

Page 12, line 9: this paragraph the author compares the RH effects of this experiments with Gaston et al. Please note that due to the difference in experimental conditions, such comparison may not be meaningful. For previous studies about heterogeneous uptake of IEPOX onto acidic particles (Gaston, Riedel et al. 2014, Riedel, Lin et al. 2015, Zhang, Chen et al. 2018), the acidic particles are in excess. But for this study, based on the mass concentration of the H2SO4 particles that the author provides, the concentration of H2SO4 particles might be limited, leading the majority of the isoprene SOA particles coming from self-nucleation. Therefore, the isoprene-OH SOA formation mechanisms between this study and past studies are quite different. The author should either provide SMPS data to show that there is no significant number concentration increase (i.e., isoprene SOA formed from heterogeneous reaction rather than self-nucleation), or revise the statement in this paragraph and point out the difference between this study and Gaston et al. Moreover, according to Gaston et al., when pH>2, the uptake coefficient is already very small. The pH of the particles in this study is estimated to be "2-3.7", which makes a comparison with Gaston et al. not meaningful.

[Figure]

Page 11, line 10: the author states "Since O−H-containing products from terpene are 10 more enriched from the OH channel oxidation than from the O3 one (Calogirou et al., 1999), here the absorption ratio of O−H to CïijİO was used to examine the difference between the O3 and OH oxidation channels." Could the author provide any rationale that why the IR results of terpene SOA could be used to infer the IR results of isoprene SOA?

Minor Comments

Page 6, lines 18-20: this part of the manuscript is a bit confusing. The author talked about the maximum mass concentrations ratios when comparing humid v.s. dry conditions for toluene-NO2 and isoprene-NO2 reactions. However, the author did not report such maximum mass concentration ratios of humid v.s. dry conditions for isoprene-O3 and isoprene-H2O2 conditions. The author should list all the ratios together for readers to understand and compare.

Page 8, line 15: please define E2/E3

Page 8, lines 20-24: This part is a bit confusing to read. First, the author uses "why is the maximum yield from isoprene-H2O2 irradiations unchanged under humid conditions." Shouldn't there be a question mark instead of a period at the end of the sentence? Then the author should provide some explanation to this question rather than just leaving the statement unattended. Next, the author talks about RH influence on aromatic SOA. At the end of this paragraph (page 9, line 1), the author changes back to isoprene SOA again. It can be difficult for readers to follow the thought process of this paragraph. I suggest the author revise this part to make it flow better.

Page 9, line 22: the author states that SOA mass concentration increased by 16%. Was this increase due to LWC, or was it actually organic mass? The author should explain it more clearly.

'Page 14, line 19: I believe the author had the statement inversed. It should be: "the

range of 300 to 800 m/z under humid conditions is reduced by 75% as compared to that under dry conditions."

Page 19, line 11: there should be a space between 5% and RH.

Page 19, line 11: there is an extra parenthesis.

References

Gaston, C. J., et al. (2014). "Reactive Uptake of an Isoprene-Derived Epoxydiol to Submicron Aerosol Particles." Environ. Sci. Technol. 48(19): 11178-11186.

Riedel, T. P., et al. (2015). "Heterogeneous Reactions of Isoprene-Derived Epoxides: Reaction Probabilities and Molar Secondary Organic Aerosol Yield Estimates." Environ. Sci. Technol. Lett. 2(2): 38-42.

Zhang, Y., et al. (2018). "Effect of Aerosol-Phase State on Secondary Organic Aerosol Formation from the Reactive Uptake of Isoprene-Derived Epoxydiols (Iepox)." Environ. Sci. Technol. Lett.

---

## Author Comment (AC1) · 23 Mar 2018

**Response to Reviewer 1**

We greatly appreciate the time and effort that reviewer 1 spent in reviewing our manuscript. The comments are really thoughtful and helpful to improve the quality of our paper. Reviewer 1 has provided both major comments and other specific comments. Below we make a point-by-point response to these comments. According to editor's requirement, the response to the referee is structured in the following sequence: (1) comments from the referee in black color, (2) our response in blue color, and (3) our changes in the revised manuscript in red color.

This manuscript on different roles of water in toluene and isoprene SOA formation presents interesting results. Overall, the results are consistent with previous studies: toluene SOA formation is enhanced by higher RH due to aqueous-phase reactions and isoprene SOA formation is enhanced by lower RH due to oligomer formation. The main finding suggests that the isoprene SOA (ozonolysis and $NO_2$ irradiation) formation is largely controlled by the "SCI → oligomer" process that is influenced by water. The manuscript is well written and provides sufficient measurements supporting the main conclusion, but a few issues need to be addressed before considered publication in ACP.

**Main Comments:**

1. On page 2, line 14, the authors claimed that "the relationship between RH and SOA formation from isoprene is still not very clear". However, from the previous studies listed in this paragraph, it appears that a pretty clear understanding has been established: (1) oligomers have always been found to enhance under low RH; and (2) SOA yield may not always enhance accordingly, because the competing pathways such as glyoxal SOA or organosulfate formation are increased under high RH and the overall SOA mass vary with the relative contributions from different pathways. Thus, it is problematic to say the RH effect on isoprene SOA is "not very clear". Instead, what remains a question is, whether the known MAE-derived oligomers explain all oligomers. The authors' findings fit right into this question and should thus set up this direction in the

introduction. Also, in the introduction, the authors should provide more details of previous work that has found SCI as key oligomeric chain unit and give more credit to these studies (e.g., Sadezky et al., 2008 ACP and a few other studies). Right now, it was only briefly mentioned that SCI-derived products have been found in SOA, but considering the results of this work, the relationship between SCI and oligomers should be elaborated in more detail.

We agree with the reviewer that the relationship between RH and SOA from isoprene is clear. However, the mechanism for the influence of RH on SOA derived from radicals is not very clear. Following the reviewer's comments, we have deleted the original sentence of line 14 on page 2: "the relationship between RH and SOA formation from isoprene is still not very clear".

We have added some sentences to introduce the important SOA precursor of MAE in the Introduction section on page 2.

MPAN is one of key precursors of SOA from isoprene under high $NO_x$ conditions (Surratt et al., 2010), which can react with OH to produce epoxides (methacrylic acid epoxide (MAE), hydroxymethyl-methyl-a-lactone (HMML)). Lin et al. (2013) reported that MAE was an important precursor to 2-MG, a tracer of isoprene-derived SOA. Nguyen et al. (2015) showed that HMML could form SOA. Since SCIs, IEPOX, MPAN, HMML and MAE co-exist in isoprene-$NO_2$ irradiations, there are cross reactions in the system. Thus, the study is still needed to demonstrate the role of these precursors in oligomer formation from isoprene-$NO_2$ irradiations.

In addition, we have added a new paragraph to review the progress on the SCI-derived oligomer in SOA.

Sadezky et al (2006, 2008) reported that SCIs ($CH_2OO$, $C_2H_4OO$, $C_3H_6OO$ and $C_4H_8OO$) play a central role in SOA formation from the ozonolysis of ethyl butenyl ether, trans-3-hexene, 2, 3-dimethyl-2-butene, and trans-4-octene. They further suggested that SCI-derived oligomers are formed by the reactions of $RO_2$ with SCI. Sakamoto et al. (2013) showed that the reactions of SCI ($CH_2OO$) with hydroperoxides (ROOH) from ethylene can form SOA. Inomata et al (2014) showed that the reaction

of an SCI with carboxylic acids or hydroperoxides can contribute to SOA formation from the ozonolysis of isoprene. Zhao et al. (2015, 2016) also showed that the SOA generated from the ozonolysis of trans-3-hexene and α-cedrene is primarily composed of oligomers formed from the addition of SCI to $RO_2$ radicals. Although these studies have demonstrated the importance of SCI-derived oligomers in SOA formation from the ozonolysis of alkenes, the role of SCI in SOA formation from isoprene-$NO_2$ irradiations has not been reported.

2. In section 3.1, the authors stated that the similar RH effect on isoprene SOA from ozonolysis and $NO_2$ irradiation experiments suggest that the ozonolysis of isoprene is a key pathway influencing the SOA formation in isoprene-$NO_2$ irradiations. This is not necessarily true. In the isoprene-NOx experimental conditions used in Zhang et al. (2011 ACP), most isoprene were reacted with OH before ozone was formed. In the Lewandowski et al. (2015 ACP) study, continuous mode was used and thus minimal ozone is expected to form. Nguygen et al. (2011 ACP) does not produce much $O_3$ and still observe enhanced oligomers under dry conditions (despite SOA yield was not affected). In these studies, ozonolysis play a small role in isoprene oxidation and observed similar results (or the authors may provide MCM simulations and prove otherwise). Even for the studied condition, as the authors stated, isoprene reacts with OH (59%) and $O_3$ (25%) and the SOA yield from the OH channel was over 2 (5) times greater than that from the $O_3$ channel under dry (humid) conditions. Thus, at most 10-15% of isoprene SOA are from the $O_3$ channel. This contradicts to the observed doubled SOA formation under dry conditions if all the RH effect is from the $O_3$ channel.

We agree with the reviewer that ozonolysis is not necessarily to be the key channel influencing SOA in isoprene-$NO_2$ systems, which obviously depends on specific experimental conditions, including extra OH sources. Indeed, our statement is based on our experimental conditions, where the yield of SCI was much higher than those of IEPOX, MPAN, HMML and MAE. Taking the reviewer's advice, the original statement that "This shows that the ozonolysis of isoprene is a key pathway influencing the SOA formation in isoprene-$NO_2$ irradiations." has been changed to

Thus, it shows that the ozonolysis of isoprene may be a key pathway influencing SOA formation in isoprene-$NO_2$ irradiations in our experimental conditions, which will be further discussed in latter section."

In these studies mentioned by the reviewer, their experimental conditions are different from our ones. For example, in the study by Zhang et al. (2011), artificial seed particles were used. In Nguygen et al. (2011 ACP) H2O2 was added. Thus, the role of ozonolysis in these studies is different from that in our study. Since there are synergistic or competing mechanisms for SOA formation when $NO_2$ and $O_3$ are both present, we cannot deduce SOA contribution simply by initial ratios of isoprene oxidized by OH and $O_3$. However, since SOA was mainly formed by the secondary or later generation products, we can evaluate SOA contributions by precursors of SOA from different channels.

As discussed in the section of Introduction in our manuscript, SCI can be taken as SOA precursor from the $O_3$ channel, while IEPOX, MPAN, HMML and MAE as SOA precursors from the OH channel. Following the reviewer's suggestion, we simulated the concentrations of IEPOX, MPAN, HMML, MAE and total yield of SCI using MCM (based on Exp. 25). Results that the total yield of SCI is dominant as compared to OH channel precursors such as IEPOX, MPAN, HMML and MAE. The former (SCI) accounts for 70% and the remaining precursors (IEPOX+MPAN+HMML+MAE) 30% at the end of reaction in isoprene-$NO_2$. Therefore, in the system of isoprene-$NO_2$, even though 59% of isoprene was consumed by OH and only 25% by $O_3$, the formation of SOA was still mainly from the $O_3$ channel.

To make it more clearly, we have following sentences on page 7:

There are cross reactions when $NO_2$ and $O_3$ are both present. Thus, we cannot deduce SOA contribution simply by initial ratios of isoprene oxidized by OH and $O_3$. Since SOA was mainly formed by the secondary or later generation products, we could evaluate the contribution of reaction pathways to formation of SOA in terms of SOA precursors from different channels. As described previously, SCI can be taken as the SOA precursor from the $O_3$ channel, while IEPOX, MPAN, HMML and MAE can be used as SOA precursors from the OH channel. The MCM simulations show that the

total yield of SCI was dominant as compared to OH channel precursor such as IEPOX, MPAN, HMML and MAE. The former accounts for 70% of SOA precursors while the latter (IEPOX+MPAN+HMML+MAE) 30% at the end of reaction in isoprene-NO$_2$. Therefore, even though 59% of isoprene were consumed by OH and only 25% by O$_3$, the formation of SOA in isoprene-NO$_2$ was mainly from the O$_3$ channel.

3. It appears that the authors used MCM simulation a few times in this study. I feel it might be worthwhile adding a section describing what they did. Particularly, SCI + RO$_2$ reactions and SCI derived oligomer formation were added by the authors in the mechanism. Quantitatively, how much could the modeled total SCI-derived oligomers account for the measured total isoprene SOA? Other references include, for example, the authors claiming MPAN+OH needs extra OH source. This is not the case in the Lin et al. (2013 PNAS). Without clearly show the MCM simulations, it is hard to draw the conclusion. These are very important aspects in the results and should be provided in more detail.

We cannot quantify the contribution from SCI-derived oligomers to isoprene SOA because many species can react with SCI, but details are not known so far. We can only simulate the reactions of SCI with glyoxylic acid and ACETOL that have been identified by our MS. The purpose of our simulation was to evaluate the role of RH in the formation of SOA. The original sentences at lines 9~ 12 on page 14 have been deleted, and a figure and some sentences have been added on page 20:

Following the reviewer's suggestion, we have added a paragraph in Experimental section to describe the model simulation on page 5:

To evaluate the potential contribution of SOA precursors (e.g. glyoxal, IEPOX, MPAN, HMML, MAE and SCI) from toluene and isoprene reaction systems, a model of the Master Chemical Mechanism (MCM v3.3.1, website: http:// mcm.leeds.ac.uk/MCM, Jenkin et al., 2015) was used, which includes the chamber dependent reactions. To examine the influence of RH on oligomer formation from SCI, the reactions of SCI

with carbonyls were added to MCM, which were expressed with $X$+SCI=$X$(SCI)$_1$, $X$(SCI)$_1$+SCI=$X$(SCI)$_2$… $X$(SCI)$_{n-1}$+SCI=$X$(SCI)$_n$, where $n$=1-10 and $X$ represents carbonyls. The rate constant for these reactions was set to be $5 \times 10^{-12}$ cm$^3$ molecule$^{-1}$ s$^{-1}$ (Vereecken et al., 2012). Since most of RO$_2$ was consumed by NO$_x$, SCI+RO$_2$ reactions were not included in our model. The carbonyls were chosen based on the results of mass spectra data from isoprene-NO$_2$ irradiations shown in section 3.4.1. A set of ordinary differential equations was built and solved using Matlab."

To quantify the contribution from SCI-derived oligomers to SOA, we have made MCM simulations. The original sentences on page 14 line 9~ 12 have been deleted, and a figure and some sentences have been added on page 18:

To quantify the RH effect of SOA and relatively possible contribution of SCI-derived oligomers from 2 species in isoprene-NO$_2$ irradiations, the reactions of SCI with formic acid, glyoxylic acid and ACETOL were added into MCM. Simulations show that the total mass concentration of oligomers from these reactions was 558.4 (271.2) μg/m$^3$ at 7% (80%) RH. The mass concentrations of SCI -derived oligomers reduced by 51% as RH increased from 7% to 80%, while the concentrations of other precursors had little change under different RH conditions. In addition, simulation also shows that the mass concentrations form other SOA precursors IEPOX, MPAN, HMML and MAE were 182.8(167.0), 27.4(28.9), 28.1(27.4), and 11.2(10.9) μg/m$^3$ at 7% (80%) RH (Figure 13). It is obvious that SCI-derived oligomers from glyoxylic acid and ACETOL should have a great potential for formation of SOA, compared to other precursors.

[Figure]

Figure 13 the MCM simulated the time profiles of SOA precursors in isoprene-NO₂ irradiation.

Indeed, Lin et al. (2013, PNAS) did not use an extra OH source in their study. We made the mistake. This has been corrected: "(Surratt et al., 2010; Lin et al., 2013; Nguyen et al., 2015) at lines 17-18, page 13" has been changed to

(Surratt et al., 2010; Nguyen et al., 2015)

4. The mass spectral data suggest "the SCI based oligomers are almost 2 times larger than that from 2-MG". However, the data are not quantitatively calibrated. It is unclear which is more important.

It is true that the MS data were not quantitatively calibrated in our study. The HR-MS data show that the SOA from isoprene-NO₂ irradiations is mainly composed of hundreds of oligomers. It is very difficult to prepare the standard substances for so many compounds.   Since the oligomers from SCI and 2-MG (or MAE) have similar structures, we considered that their peak heights in MS reflect equivalent mass. Thus, the total peak heights from SCI-based oligomers were used to compare with those from 2-MG in our work.

Overall, the authors reporting isoprene SCI-derived oligomers in SOA is well justified and important. These oligomers can be important fractions of oligomers under isoprene high-NOx conditions and might explain previously observed oligomers in isoprene SOA besides those derived from MAE. However, based on the results provided in this work, it is not convincing that the SCI-derived oligomers are the dominant part in isoprene SOA, especially under isoprene-$NO_2$ irradiation conditions.

We have pointed out in our revised manuscript that our conclusion about important ozonolysis influencing SOA formation in isoprene-$NO_2$ systems was made under our experimental conditions, which is not only based on the yields between isoprene-$NO_2$ with isoprene-$O_3$ under dry and humid conditions, but also on the following facts: (1) The FTIR spectra of SOA from isoprene-$NO_2$ irradiations show that the absorbance ratios of O−H/C═O are 0.35 (0.36) under dry (humid) conditions, which are almost the same as the corresponding values in isoprene-$O_3$ but totally different from the values in isoprene-$H_2O_2$ (the ratio are 1.63 (dry) and 1.45 (humid)). In addition, the similar features of IR spectra between isoprene-$NO_2$ and isoprene-$O_3$ reaction systems also demonstrate that the $O_3$ channel plays an import role in isoprene-$NO_2$ irradiations. (2)The mass spectrum of SOA from the isoprene-$NO_2$ system is similar to the one from the ozonolysis of isoprene, which all shows obviously regular structures of the peaks for oligomers with SCI as a base unit; while the spectrum of SOA from isoprene-$H_2O_2$ does not such regular structures, which shows a feature different from that of the SOA from isoprene-$NO_2$. (3) The MCM simulation further demonstrates the importance of the SCI-derived oligomers in the formation of isoprene SOA under our experimental conditions.

Minor comment:
1. Page 2, line 16. The term "isoprene-NOx (x=1, 2)" is odd. Suggest using "isoprene-NOx (NO and $NO_2$)".
We have corrected it.

2. Page 2, line 19. Nguyen et al. (2011) did not find a RH influence on SOA yield, but did observe enhancement of oligomers under low RH, which is consistent with Zhang et al. This should be mentioned in this sentence.

We have added following words "but they observed enhancement of 2-MG -derived oligomers under low RH, which was consistent with Zhang et al. (2011a)"

3. Figures. The 4 plots in Figure 1 need to be numbered. The authors have consistently used red colors for low-RH results and black for high-RH results in many figures which is very helpful. I suggest use the same color scheme in Figures 6 (replace the blue line with a black dashed line), 10 (use colors for symbols and lines), 11, and 12.

It is very good suggestions! We have numbered plots in Figure 1, and changed the color in Figure 6, 10, 11 and 12.

4. Page 7, line 3. There are other possible reasons accounting for the lower SOA yield. For example, this study uses higher temperature than the two previous studies mentioned.

We have added following words: "In addition, the temperature in this study is higher than the previous studies, which may be another reason accounting for the lower SOA yields in this work"

5. Page 8, line 5. Is this based on MCM simulation? Please add clarification in the text.

Yes, it is based on MCM simulation, to clarify it, we have added the following words to the original sentence: "based on MCM simulation"

6. Page 13, line 17. "MPAN can be oxidized by OH to form 2-methyltetrols" is wrong. The products are 2-methylglyceric acid and related oligomers. This is repeated by later sentences in this paragraph. The authors should consider consolidate/rewrite this paragraph.

We have rewritten the paragraph:

The yield of MACR is generally greater in isoprene-$NO_2$ irradiations and isoprene-$O_3$ systems than that in isoprene-$H_2O_2$ irradiations. MACR can react to form MPAN in the presence of $NO_2$, which can be oxidized by OH to form SOA precursors of epoxides (e.g., HMML, MAE), such as in the Nguyen et al. (2011) work with $H_2O_2$ as an extra OH source. Epoxides can further be oxidized to produce 2-MG and related oligomers (Surratt et al., 2010; Lin et al., 2013; Nguyen et al., 2015). 2-MG-derived oligomers can be enhanced under lower RH (Zhang et al., 2011). Both the results from Nguyen et al. (2014) and MCM simulations further show that if there are enough OH radicals, most of MPAN can be further oxidized by OH to produce epoxides. However, since there were no extra OH sources in our systems, MCM simulations show that only 12% (24%) of MPAN under dry (humid) conditions was oxidized by OH to produce HMML and MAE. The maximum concentrations of HMML and MAE were only 6.8 and 2.7 ppb under dry conditions (Figure 4), which is too small to explain the yields of SOA in isoprene-$NO_2$ systems."

7. Page 14, line 6. "Thus, IEPOX is not the major contributor to SOA in isoprene-$NO_2$ system. This clearly demonstrates that MPAN and IEPOX related reactions were not dominant pathways for SOA formation in our isoprene-$NO_2$ irradiations". The logic in this sentence is problematic. It cannot suggest MPAN reactions are not important in isoprene-$NO_2$ irradiations. Again, a figure showing simulated MPAN, SCI, and a few other important products is needed at least.

The logic of the original sentence is indeed problematic. Following the reviewer's comments, the original sentence "This clearly demonstrates that MPAN and IEPOX related reactions were not dominant pathways for SOA formation in our isoprene-$NO_2$ irradiations." has been deleted. As our response to main comment 2, we believe that MPAN has a small contribution to SOA in our reaction system because the reacted MPAN is low in our experimental conditions. In addition, based on data from FTIR and MS spectra we suggest that SCI-based oligomer are major components of SOA in isoprene-$NO_2$ irradiations. We have added a figure (Figure 4) to show the concentrations of the products in our system.

8. Page 18, line 9. C4H6O3 is not formed from dehydration of 2-MG, but MAE (Lin et al., 2013 PNAS).

Taking the reviewer's comment, we have added the following sentences on page 18:

Lin et al. (2013) has reported that $C_4H_6O_3$ was from MAE in MACR-$NO_x$ irradiations. Thus, $C_4H_6O_3$ is probably formed from dehydration of 2-MG and MAE in oligomers. Considering the low yield of MAE in our system, we considered that most of $C_4H_6O_3$-based oligomers is probably contributed by 2-MG in our work.

---

## Author Comment (AC2) · 23 Mar 2018

**Response to Reviewer 2**

We greatly appreciate the time and effort that reviewer 2 spent in reviewing our manuscript. The comments are really thoughtful and helpful to improve the quality of our paper. Reviewer 2 has provided both major comments and other specific comments. Below we make a point-by-point response to these comments. According to editor's requirement, the response to the referee is structured in the following sequence: (1) comments from the referee in black color, (2) our response in blue color, and (3) our changes in the revised manuscript in red color.

The manuscript is comprehensive and provides useful information about the chemical composition of SOA formed from various pathways. However, there are a few points that the author should consider when making these arguments, and these issues should be addressed before the article is published in ACP.

**Major Comments**

First, the author draws the conclusion that RH does not have an effect on isoprene SOA yield under the OH oxidation pathway. However, Gaston et al. (2014) clearly show that RH has an effect on isoprene epoxydiols (IEPOX) reactive uptake into the acidic sulfate particles due to the dilution effect at higher RH. Because IEPOX is one of the main oxidation products of isoprene under OH pathway, the yield should show a negative correlation with RH. But the author's experimental results seem to not agree with those of past studies. I believe the main reason for non-RH dependence observed in this study is that the acidity of the $H_2SO_4$ particles generated by this study is too low, resulting in RH not having an additional acidity dilution effect. The author should at least mention the discrepancy between this work and other studies, and point out limitations of this study. For instance, page 7, line 19-20: the author made a conclusion that RH had little effect on the yield of isoprene with OH reactions. Please note that this is probably true for generating self-nucleated isoprene SOA, but may not be true for isoprene SOA that are formed on acidic sulfate seed particles, as shown by a few other studies (Gaston,

Riedel et al. 2014, Riedel, Lin et al. 2015, Zhang, Chen et al. 2018). These studies suggest that RH has an important effect on the formation of isoprene OH-heterogeneous generated SOA, due to the change of RH affecting the acidity of the seed particles. The author should include the literature and clarify the difference of RH effects on experiments because the seed particles in this study probably do not have enough concentration to undergo full heterogeneous reactions.

We agree with the reviewer that there is discrepancy between our work and other studies. The initial burst of new particles happened at 10 min after the start of irradiation in isoprene-$H_2O_2$ system in our work and maximum number concentration was over $10^4$. These new particles were considered to be $H_2SO_4$ particles. Both theoretical simulation and experiments have confirmed the formation of these new particles. We ever did the experiment. If the $SO_2$ concentration in background air was removed to be below 50 ppt, the number concentration of $H_2SO_4$ would be below 1000 /$cm^3$. In addition, using the extra-purified background air, the mass concentration of SOA would be greatly reduced by one order of magnitude in isoprene-OH system. We have added the curve in the revised Figure 2 to show the variation of number concentrations of particle with time from the experiment of ISO-$H_2O_2$, indicating the formation of $H_2SO_4$ and heterogeneous generated SOA. As the reviewer pointed out, the mass concentration of $H_2SO_4$ was much smaller as compared to previous studies. Although the heterogeneous formation of SOA occurred in our system, compared with other studies the acid dilution effect was not obvious under higher RH condition. To make it clear, we have added the following sentences on page 7 and references to clarify the differences between our work and previous studies.

[Figure]

Figure 1 Figure 2 in our manuscript

In addition, some other previous studies (Gaston et al., 2014; Riedel et al. 2015; Zhang et al. 2018) showed that RH has a negative effect on the formation of SOA from isoprene-OH systems due to acid dilution effect. In these studies, acidic sulfate seed particles were used and the acid-catalyzed effect was very obvious. Thus, higher RH can reduce the acidity of the seed particles by particle water. In our study acidic seed particles were from a little amount of $H_2SO_4$ formed from the gas-phase reaction of $SO_2$ and OH. It was estimated that the mass concentration of $H_2SO_4$ particles was less than 1 μg/m$^3$. When liquid water content increased from 1 μg/m$^3$ to the maximum 54 μg/m$^3$ under humid conditions, the pH value was estimated to be in the range of 2 to 3.7, indicating that the pH variation was small in our experimental conditions. Therefore, compared with previous studies, the acid dilution effect was not remarkable in our work.

Riedel, T. P., Lin, Y. H., Budisulistiorini, S. H., Gaston, C. J., Thornton, J. A., Zhang, Z., Vizuete, W., Gold, A. and Surratt, J. D.: Heterogeneous reactions of isoprene-derived epoxides: Reaction probabilities and molar secondary organic aerosol yield estimates, Environ. Sci. Technol. Lett., 2(2), 38–42, doi:10.1021/ez500406f, 2015.

Zhang, Y., Chen, Y., Lambe, A. T., Olson, N. E., Lei, Z., Craig, R. L., Zhang, Z.,

Gold, A., Onasch, T. B., Jayne, J. T., Worsnop, D. R., Gaston, C. J., Thornton, J. A., Vizuete, W., Ault, A. P. and Surratt, J. D.: Effect of the Aerosol-Phase State on Secondary Organic Aerosol Formation from the Reactive Uptake of Isoprene-Derived Epoxydiols (IEPOX), Environ. Sci. Technol. Lett., 5(3), acs.estlett.8b00044, doi:10.1021/acs.estlett.8b00044, 2018.

Page 7, lines 10-11: the author concludes that ozonolysis is the key pathway influencing SOA formation in isoprene-$NO_2$ systems. How does the author draw such conclusion? It seems that the author just compares the yields of isoprene-$NO_2$ with isoprene-$O_3$ under dry and humid conditions, and finds out the yields were similar. But similar yields do not mean that the reaction mechanisms are similar. The ozone concentrations in the two conditions could be different. There might also be synergistic or competing mechanisms for SOA formation when $NO_2$ and $O_3$ are both present. Besides, the author also shows on page 7, lines 21-22 that $NO_2$ and $O_3$ pathway each contributes to isoprene oxidation for certain extent. Therefore, the author cannot rule out the possibility that $NO_2$ and $O_3$ pathways having similar yields, rather than $O_3$ pathway dominates the $NO_2$ pathway. I think the author should either provide more evidence in this part to show isoprene-$O_3$ really dominates, or make a more conservative the conclusion and state the limitations.

The reviewer is absolutely right. We can't draw this conclusion only based on the comparison of SOA yields between isoprene-$NO_2$ and isoprene-$O_3$ systems. Actually, our conclusion was based on mass yields, chemical information provided by FTIR and HR-MS data, and MCM simulations.

Just as reviewer pointed out, there are synergistic or competing mechanisms for SOA formation when $NO_2$ and $O_3$ are both present. Since SOA was mainly formed by the secondary or later generation products, we could evaluate the contributions of SOA precursors from different channel. As discussed in the section of Introduction in our manuscript, SCI can be taken as SOA precursor from the $O_3$ channel, while IEPOX,

MPAN, HMML and MAE as SOA precursors from the OH channel. The MCM model simulation (based on Exp. 25) showed that the total yield of SCI is dominant as compared to OH channel precursors such as IEPOX, MPAN, HMML and MAE. The former (SCI) accounts for 70% and the remaining precursors (IEPOX+MPAN+HMML+MAE) 30% at the end of reaction in isoprene-$NO_2$. Therefore, in the system of isoprene-$NO_2$, even though 59% of isoprene was consumed by OH and only 25% by $O_3$, the formation of SOA was still mainly from the $O_3$ channel.

Following the reviewer's suggestion, the original sentence of Page 7, lines 10-11 has been changed as:

Thus, these results show that high RH can reduce the maximum yields of SOA from the reaction channel of isoprene with $O_3$ ($O_3$ channel) and that RH has little effect on the maximum yields from the reaction channel of isoprene with OH (OH channel) without sufficiently high mass concentrations of acid particles. This shows that the ozonolysis of isoprene is probably a key pathway influencing the SOA formation in isoprene-$NO_2$ irradiations.

To make it clear, we have added the following sentences on page 7:

There are cross reactions when $NO_2$ and $O_3$ are both present. Thus, we cannot deduce SOA contribution simply by initial ratios of isoprene oxidized by OH and $O_3$. Since SOA was mainly formed by the secondary or later generation products, we could evaluate the contribution of reaction pathways to formation of SOA in terms of SOA precursors from different channels. As described previously, SCI can be taken as the SOA precursor from the $O_3$ channel, while IEPOX, MPAN, HMML and MAE can be used as SOA precursors from the OH channel. The MCM simulations show that the total yield of SCI was dominant as compared to OH channel precursor such as IEPOX, MPAN, HMML and MAE. The former accounts for 70% of SOA precursors while the latter (IEPOX+MPAN+HMML+MAE) 30% at the end of reaction in isoprene-$NO_2$. Therefore, even though 59% of isoprene were consumed by OH and only 25% by $O_3$, the formation of SOA in isoprene-$NO_2$ was mainly from the $O_3$ channel.

Page 7, line 23: the author points out that RH has little effects on $NO_3$ pathway. Please

list any evidence (such as references) to support this point.

We have added 3 references and the following sentence to support our statement on page 7. Previous studies have shown that humid has little effect on SOA formation from NO$_3$ oxidation of alkenes (Bonn and Moorgat 2002; Fly et al., 2009; Boyd et al., 2015).

Boyd, C. M., Sanchez, J., Xu, L., Eugene, A. J., Nah, T., Tuet, W. Y., Guzman, M. I. and Ng, N. L.: Secondary organic aerosol formation from the β-pinene+NO$_3$ system: Effect of humidity and peroxy radical fate, Atmos. Chem. Phys., 15(13), 7497–7522, doi:10.5194/acp-15-7497-2015, 2015.

Bonn, B. and Moorgat, G. K.: New particle formation during α- and β-pinene oxidation by O$_3$, OH and NO$_3$, and the influence of water vapour: particle size distribution studies, Atmos. Chem. Phys., 2(3), 183–196, doi:10.5194/acp-2-183-2002, 2002.

Fry, J. L., Rollins, a W., Wooldridge, P. J., Brown, S. S., Fuchs, H. and Dub, W.: Organic nitrate and secondary organic aerosol yield from NO$_3$ oxidation of β-pinene evaluated using a gas-phase kinetics/aerosol partitioning model, Atmos. Chem. Phys., 9(3), 1431–1449, doi:10.5194/acp-9-1431-2009, 2009.

Page 12, line 9: this paragraph the author compares the RH effects of this experiments with Gaston et al. Please note that due to the difference in experimental conditions, such comparison may not be meaningful. For previous studies about heterogeneous uptake of IEPOX onto acidic particles (Gaston, Riedel et al. 2014, Riedel, Lin et al. 2015, Zhang, Chen et al. 2018), the acidic particles are in excess. But for this study, based on the mass concentration of the H$_2$SO$_4$ particles that the author provides, the concentration of H$_2$SO$_4$ particles might be limited, leading the majority of the isoprene SOA particles coming from self-nucleation. Therefore, the isoprene-OH SOA formation mechanisms between this study and past studies are quite different. The author should either provide SMPS data to show that there is no significant number concentration increase (i.e., isoprene SOA formed from heterogeneous reaction rather

than self-nucleation), or revise the statement in this paragraph and point out the difference between this study and Gaston et al. Moreover, according to Gaston et al., when pH>2, the uptake coefficient is already very small. The pH of the particles in this study is estimated to be "2-3.7", which makes a comparison with Gaston et al. not meaningful.

We totally agree with the reviewer that our experimental conditions are different from those by previous studies. As discussed in the response to the first comment by the reviewer, though the mass concentration of acid particles was not high in our system, it is considered that SOA was still mainly formed by heterogeneous reaction in our isoprene-OH systems. To avoid needless repetitions, the original paragraph of page 12, lines 9-18 has been deleted.

Page 11, line 10: the author states "Since O-H-containing products from terpene are 10 more enriched from the OH channel oxidation than from the $O_3$ one (Calogirou et al., 1999), here the absorption ratio of O-H to C=O was used to examine the difference between the $O_3$ and OH oxidation channels." Could the author provide any rationale that why the IR results of terpene SOA could be used to infer the IR results of isoprene SOA?

Considering isoprene is the basic chain unit of terpenes, similar abundance of functional groups in oxidation products from isoprene and terpenes can be expected. In fact, we also measured the IR spectra of SOA from α-pinene oxidized by $O_3$ and OH (see below Figure ). The similar characteristics of IR spectra of SOA from both isoprene and α-pinene was observed. The peak height ratio of O−H/C＝O is 0.24 in SOA from α-pinene-$O_3$ system, while it is as high as 2.19 in α-pinene- OH system. This demonstrates that the absorption ratios of O-H to C=O from α-pinene are similar to SOA from isoprene. To make it clear, we have modified original sentences of page 11, lines 10-13.

[Figure]

The IR spectra of SOA from α-pinene-O₃ and α-pinene-OH systems under dry conditions.

Since isoprene is the chain unit of terpenes, the abundance of functional groups in oxidation products from isoprene and terpenes is expected to be close. O-H-containing products from terpene are 10 more enriched from the OH channel oxidation than from the O₃ one (Calogirou et al., 1999). Our extra experiments show the similar characteristics of IR spectra of SOA from both isoprene and α-pinene (figure not shown). Here the absorption ratio of O-H to C=O was used to examine the difference between the O₃ and OH oxidation channels. Higher values of O−H/C=O ratio should be expected to be more from the OH channel than from the O₃ channel. Our experiment shows that the peak height ratio of O−H/C=O is 0.24 in SOA from α-pinene-O₃ system, while it is as high as 2.19 in α-pinene-OH system.

**Minor Comments**

Page 6, lines 18-20: this part of the manuscript is a bit confusing. The author talked about the maximum mass concentrations ratios when comparing humid v.s. dry conditions for toluene-NO₂ and isoprene-NO₂ reactions. However, the author did not report such maximum mass concentration ratios of humid v.s. dry conditions for isoprene-O₃ and isoprene-H₂O₂ conditions. The author should list all the ratios together for readers to understand and compare.

We have added the maximum mass ratios for isoprene-$O_3$ and isoprene-$H_2O_2$ on page 6.

The ratio is 0.45 for isoprene-$O_3$ and 1.01 for isoprene-$H_2O_2$.

Page 8, line 15: please define E2/E3

We have added following words to define E2/E3 on page 8:

(absorbance at 250 nm divided by absorbance at 365 nm)

Page 8, lines 20-24: This part is a bit confusing to read. First, the author uses "why is the maximum yield from isoprene-$H_2O_2$ irradiations unchanged under humid conditions." Shouldn't there be a question mark instead of a period at the end of the sentence? Then the author should provide some explanation to this question rather than just leaving the statement unattended. Next, the author talks about RH influence on aromatic SOA. At the end of this paragraph (page 9, line 1), the author changes back to isoprene SOA again. It can be difficult for readers to follow the thought process of this paragraph. I suggest the author revise this part to make it flow better.

We thanks the reviewer for the careful reading and good suggestions. According to the reviewer's suggestions, we have added the question mark at the end of the sentence, and also briefly explained it. Further discussion is given in later sections. This part has been revised on page 9.

Nevertheless, if the suppression of the oligomerization reactions under humid conditions is the main reason for the decrease in SOA yield from isoprene, why is the maximum yield from isoprene-$H_2O_2$ irradiations unchanged under humid conditions? Besides the weak acid dilution effect in our experimental conditions, there must be an intrinsic mechanism regarding the influences of RH on the SOA yield from isoprene. In addition, oligomers have been also identified as important products of SOA from aromatics, and water is a byproduct during oligomerization process (Lim et al., 2010; Gaston et al., 2014; Kalberer et al., 2004). However, a negative effect of RH on SOA yield from aromatics has never been observed. This is because there are likely competing processes that are responsible for SOA formation from aromatics under

humid conditions. Oligomers are generally inhibited by higher RH, while the organics formed by aqueous reactions are enhanced.

Page 9, line 22: the author states that SOA mass concentration increased by 16%. Was this increase due to LWC, or was it actually organic mass? The author should explain it more clearly.

It was organic mass. To make it clear, the original sentence has been modeified on page 9.

The particles were mainly consistent of $NaNO_3$, LWC and SOA at the end of reaction. the major changes for SOA were all the bands assigned to O−H, C＝O and C−OH (错误!未找到引用源。e), which were enhanced by 50%, 29% and 35% respectively. After considering the mass contributions from $NaNO_3$ and LWC, the organic mass concentrations of SOA were increased by 16%, compared to those in humid conditions without extra LWC.

'Page 14, line 19: I believe the author had the statement inversed. It should be: "the range of 300 to 800 m/z under humid conditions is reduced by 75% as compared to that under dry conditions."

The reviewer is right. We had the statement inversed. It has been corrected in our revised manuscript.

Page 19, line 11: there should be a space between 5% and RH.

It has been corrected.

Page 19, line 11: there is an extra parenthesis.

The extra parenthesis has been deleted.

---

## Author Response (AR1)

**Response to Reviewer 1**

We greatly appreciate the time and effort that reviewer 1 spent in reviewing our manuscript. The comments are really thoughtful and helpful to improve the quality of our paper. Reviewer 1 has provided both main comments and other specific comments.

5 Below we make a point-by-point response to these comments. According to editor's requirement, the response to the referee is structured in the following sequence: (1) comments from the referee in black color, (2) our response in blue color, and (3) our changes in the revised manuscript in red color.

This manuscript on different roles of water in toluene and isoprene SOA formation presents interesting results. Overall, the results are consistent with previous studies: toluene SOA formation is enhanced by higher RH due to aqueous-phase reactions and isoprene SOA formation is enhanced by lower RH due to oligomer formation. The main finding suggests that the isoprene SOA (ozonolysis and NO2 irradiation) formation is largely controlled by the "SCI  $\rightarrow$  oligomer" process that is influenced by water. The manuscript is well written and provides sufficient measurements supporting the main conclusion, but a few issues need to be addressed before considered publication in ACP.

15

10

**Main Comments:**

On page 2, line 14, the authors claimed that "the relationship between RH and SOA formation from isoprene is still not very clear". However, from the previous studies listed in this paragraph, it appears that a pretty clear understanding has been established: (1) oligomers have always been found to enhance under low RH; and (2) SOA yield may not always enhance accordingly, because the competing pathways such as glyoxal SOA or organosulfate formation are increased under high RH and the overall SOA mass vary with the relative contributions from different pathways. Thus, it is problematic to say the RH effect on isoprene SOA is "not very clear". Instead, what remains a question is, whether the known MAE-derived oligomers explain all oligomers. The authors' findings fit right into this question and should thus set up this direction in the introduction. Also, in the introduction, the authors should provide more details of previous work that has found SCI as key oligomeric chain unit and give more credit to these studies (e.g., Sadezky et al., 2008 ACP and a few other studies). Right now, it was only

briefly mentioned that SCI-derived products have been found in SOA, but considering the results of this work, the relationship between SCI and oligomers should be elaborated in more detail.

We agree with the reviewer that the relationship between RH and SOA from isoprene is clear. However, the mechanism for the influence of RH on SOA derived from radicals is not very clear. Following the reviewer's comments, we have deleted the

5 sentence in the original manuscript at line 14 on page 2: "the relationship between RH and SOA formation from isoprene is still not very clear".

We have added some sentences to introduce the important SOA precursor of MAE in the Introduction section on page 2. MPAN is one of key precursors of SOA from isoprene under high  $NO_x$  conditions (Surratt et al., 2010), which can react with OH to produce epoxides (methacrylic acid epoxide (MAE), hydroxymethyl-methyl-a-lactone (HMML)). Lin et al. (2013)

- 10 reported that MAE was an important precursor to 2-MG, a tracer of isoprene-derived SOA. Nguyen et al. (2015) showed that HMML could form SOA. Since SCIs, IEPOX, MPAN, HMML and MAE co-exist in isoprene-NO2 irradiations, there are cross reactions in the system. Thus, the study is still needed to demonstrate the role of these precursors in oligomer formation from isoprene-NO2 irradiations.
- In addition, we have added a new paragraph to review the progress on the SCI-derived oligomer in SOA on page 2. Sadezky et al. (2006, 2008) reported that SCIs (CH2OO, C2H4OO, C3H6OO and C4H8OO) play a central role in SOA formation from the ozonolysis of ethyl butenyl ether, trans-3-hexene, 2, 3-dimethyl-2-butene, and trans-4-octene. They further suggested that SCI-derived oligomers are formed by the reactions of RO2 with SCI. Sakamoto et al. (2013) showed that the reactions of SCI (CH2OO) with hydroperoxides (ROOH) from ethylene can form SOA. Inomata et al. (2014) showed that the reaction of
- 20 an SCI with carboxylic acids or hydroperoxides can contribute to SOA formation from the ozonolysis of isoprene. Zhao et al. (2015, 2016) also showed that the SOA generated from the ozonolysis of trans-3-hexene and  $\alpha$ -cedrene is primarily composed of oligomers formed from the addition of SCI to RO2 radicals. Although these studies have demonstrated the importance of SCI-derived oligomers in SOA formation from the ozonolysis of alkenes, the role of SCIs in SOA formation from isoprene-NO2 irradiations has not been reported.

2. In section 3.1, the authors stated that the similar RH effect on isoprene SOA from ozonolysis and  $NO_2$  irradiation experiments suggest that the ozonolysis of isoprene is a key pathway influencing the SOA formation in isoprene- $NO_2$  irradiations. This is not necessarily true. In the isoprene-NOx experimental conditions used in Zhang et al. (2011 ACP), most isoprene were reacted

- 5 with OH before ozone was formed. In the Lewandowski et al. (2015 ACP) study, continuous mode was used and thus minimal ozone is expected to form. Nguygen et al. (2011 ACP) does not produce much O3 and still observe enhanced oligomers under dry conditions (despite SOA yield was not affected). In these studies, ozonolysis play a small role in isoprene oxidation and observed similar results (or the authors may provide MCM simulations and prove otherwise). Even for the studied condition, as the authors stated, isoprene reacts with OH (59%) and O3 (25%) and the SOA yield from the OH channel was over 2 (5)
- 10 times greater than that from the  $O_3$  channel under dry (humid) conditions. Thus, at most 10-15% of isoprene SOA are from the  $O_3$  channel. This contradicts to the observed doubled SOA formation under dry conditions if all the RH effect is from the  $O_3$  channel.

We agree with the reviewer that ozonolysis is not necessarily to be the key channel influencing SOA in isoprene- $NO_2$  systems, which obviously depends on specific experimental conditions, including extra OH sources. Indeed, our statement is based on

15 our experimental conditions, where the yield of SCI was much higher than those of IEPOX, MPAN, HMML and MAE. Taking the reviewer's advice, the original statement that "This shows that the ozonolysis of isoprene is a key pathway influencing the SOA formation in isoprene-NO2 irradiations." has been changed to:

Thus, it shows that the ozonolysis of isoprene is probably a key pathway influencing SOA formation in isoprene- $NO_2$  irradiations in our experimental conditions, which will be further discussed in later section.

20 In the studies mentioned by the reviewer, their experimental conditions are different from ours. For example, in the study by Zhang et al. (2011), artificial seed particles were used. In Nguygen et al. (2011 ACP)  $H_2O_2$  was added. Thus, the role of ozonolysis in these studies is different from that in our study. Since there are synergistic or competing mechanisms for SOA formation when NO2 and O3 are both present, we cannot deduce SOA contribution simply by initial ratios of isoprene oxidized by OH and O3. However, since SOA is mainly formed by the secondary or later generation products, we can evaluate SOA

contributions by the precursors of SOA from different channels.

5

As discussed in the section of Introduction in our manuscript, SCIs can be taken as the SOA precursor from the O3 channel, while IEPOX, MPAN, HMML and MAE as SOA precursors from the OH channel. Following the reviewer's suggestion, we simulated the concentrations of IEPOX, MPAN, HMML, MAE and total yield of SCIs using MCM (based on Exp. 25). Results show that the total yield of SCIs is dominant as compared to OH channel precursors such as IEPOX, MPAN, HMML and MAE. The former (SCIs) accounts for 70% of total concentrations (ppb) of SOA precursors and the remaining precursors (IEPOX+MPAN+HMML+MAE) 30% at the end of reaction in isoprene-NO2. Therefore, in the system of isoprene-NO2, even though 59% of isoprene was consumed by OH and only 25% by O3, the formation of SOA was still mainly from the O3 channel. To make it more clearly, we have added the following sentences on page 7:

- 10 There are cross reactions when NO2 and O3 are both present. Thus, we cannot deduce SOA contribution simply by initial ratios of isoprene oxidized by OH and O3. Since SOA is mainly formed by the secondary or later generation products, we can evaluate the contribution of reaction pathways to formation of SOA in terms of SOA precursors from different channels. As described previously, SCIs can be taken as the SOA precursors from the O3 channel, while IEPOX, MPAN, HMML and MAE can be used as SOA precursors from the OH channel. The MCM simulations show that the total yield of SCIs was dominant as compared to OH channel precursors such as IEPOX, MPAN, HMML and MAE. The former accounts for 70% of total concentrations (ppb) of SOA precursors while the latter (IEPOX+MPAN+HMML+MAE) 30% at the end of reaction in isoprene-NO2 irradiations. Therefore, even though 59% of isoprene was consumed by OH and only 25% by O3, the formation
  - of SOA in isoprene-NO $_2$  was mainly from the O $_3$  channel.
- 3. It appears that the authors used MCM simulation a few times in this study. I feel it might be worthwhile adding a section describing what they did. Particularly, SCI + RO2 reactions and SCI derived oligomer formation were added by the authors in the mechanism. Quantitatively, how much could the modeled total SCI-derived oligomers account for the measured total isoprene SOA? Other references include, for example, the authors claiming MPAN+OH needs extra OH source. This is not the case in the Lin et al. (2013 PNAS). Without clearly show the MCM simulations, it is hard to draw the conclusion. These

are very important aspects in the results and should be provided in more detail.

We cannot quantify the contribution from SCI-derived oligomers to isoprene SOA because many species can react with SCIs, but details are not known so far. We can only simulate the reactions of SCIs with glyoxylic acid and ACETOL that have been identified by our MS.

5 Following the reviewer's suggestion, we have added a paragraph in Experimental section to describe the model simulation on page 5:

To evaluate the potential contribution of SOA precursors (e.g. glyoxal, IEPOX, MPAN, HMML, MAE and SCI) from toluene and isoprene reaction systems, a model of the Master Chemical Mechanism (MCM v3.3.1, website: http:// mcm.leeds.ac.uk/MCM, Jenkin et al., 2015) was used, which includes the chamber dependent reactions. To examine the influence of RH on oligomer formation from SCI, the reactions of SCI with carbonyls were added to MCM, which were expressed with *X*+SCI=*X*(SCI)1, *X*(SCI)1+SCI=*X*(SCI)2... *X*(SCI)*n*-1+SCI=*X*(SCI)*n*, where *n*=1-10 and *X* represents carbonyls. The rate constant for these reactions was set to be 5 × 10-12 cm3 molecule-1 s-1 (Vereecken et al., 2012). Since most of RO2 was consumed by NOx, SCI+RO2 reactions were not included in our model. The carbonyls were chosen based on the results of mass spectra data from isoprene-NO2 irradiations shown in section 3.4.1. A set of ordinary differential equations was built and

15 solved using Matlab.

The purpose of our simulation was to evaluate the role of RH in the formation of SOA. The original sentences at lines 9~ 12 on page 14 have been deleted, and a figure and some sentences have been added on page 20:

To quantify the RH effect of SOA and relatively possible contribution of SCI-derived oligomers from 2 species in isoprene-NO2 irradiations, the reactions of SCIs with formic acid, glyoxylic acid and ACETOL were added into MCM. Simulations

20 show that the total mass concentration of oligomers from these reactions was 558.4 (271.2) μg/m3 at 7% (80%) RH, and the mass concentrations form other SOA precursors IEPOX, MPAN, HMML and MAE were 182.8 (167.0), 27.4 (28.9), 28.1 (27.4), and 11.2 (10.9) μg/m3 at 7% (80%) RH (Figure 13). It is obvious that the mass concentrations of SCI-derived oligomers reduced by 51% as RH increased from 7% to 80%, while the concentrations of other precursors had little change under different RH conditions. Thus, SCI-derived oligomers from glyoxylic acid and ACETOL should have a great potential for formation of

Figure 13: MCM-simulated time profiles of SOA precursors in isoprene-NO2 irradiations. SCI-derived oligomers were from the reactions of SCI with glyoxylic acid and ACETOL.

Indeed, Lin et al. (2013, PNAS) did not use an extra OH source in their study. We made the mistake. This has been corrected: "(Surratt et al., 2010; Lin et al., 2013; Nguyen et al., 2015) at lines 17-18, page 13" has been changed to (Surratt et al., 2010; Nguyen et al., 2015)

4. The mass spectral data suggest "the SCI based oligomers are almost 2 times larger than that from 2-MG". However, the data are not quantitatively calibrated. It is unclear which is more important.

It is true that the MS data were not quantitatively calibrated in our study. The HR-MS data show that the SOA from isoprene- $NO_2$  irradiations is mainly composed of hundreds of oligomers. It is very difficult to prepare the standard substances for so many compounds. Since the oligomers from SCI and 2-MG (or MAE) have similar structures, we considered that their peak

heights in MS can reflect their corresponding mass. Thus, the total peak heights from SCI-based oligomers were used to compare with those from 2-MG in our work.

Overall, the authors reporting isoprene SCI-derived oligomers in SOA is well justified and important. These oligomers can be

- 5 important fractions of oligomers under isoprene high-NOx conditions and might explain previously observed oligomers in isoprene SOA besides those derived from MAE. However, based on the results provided in this work, it is not convincing that the SCI-derived oligomers are the dominant part in isoprene SOA, especially under isoprene-NO2 irradiation conditions.
- We have pointed out in our revised manuscript that our conclusion about important ozonolysis influencing SOA formation in isoprene-NO2 systems was made under our experimental conditions, which is not only based on the comparison of the yields between isoprene-NO2 and isoprene-O3 under dry and humid conditions, but also on the following facts: (1) The FTIR spectra of SOA from isoprene-NO2 irradiations show that the absorbance ratios of O-H/C=O are 0.35 (0.36) under dry (humid) conditions, which are almost the same as the corresponding values in isoprene-O3 but totally different from the values in isoprene-H2O2 (the ratios of 1.63 (1.45)). In addition, the similar features of IR spectra between isoprene-NO2 and isoprene-15 O3 reaction systems also demonstrate that the O3 channel plays an important role in isoprene-NO2 irradiations. (2) The mass spectrum of SOA from the isoprene-NO2 system is similar to that from the ozonolysis of isoprene, which shows obviously regular structures of the peaks for oligomers with SCI as a base unit; while the spectrum of SOA from isoprene-H2O2 (does not have such regular structures, which shows a feature different from that of the SOA from isoprene-NO2. (3) The MCM simulation further demonstrates the importance of the SCI-derived oligomers in the formation of isoprene SOA under our
- 20 experimental conditions.

**Minor comment:**

Page 2, line 16. The term "isoprene-NOx (x=1, 2)" is odd. Suggest using "isoprene-NOx (NO and NO2)".
 We have corrected it.

2. Page 2, line 19. Nguyen et al. (2011) did not find a RH influence on SOA yield, but did observe enhancement of oligomers under low RH, which is consistent with Zhang et al. This should be mentioned in this sentence.We have added the following words in this sentence: but they observed enhancement of 2-MG -derived oligomers under low

5 RH, which is consistent with Zhang et al. (2011a).

3. Figures. The 4 plots in Figure 1 need to be numbered. The authors have consistently used red colors for low-RH results and black for high-RH results in many figures which is very helpful. I suggest use the same color scheme in Figures 6 (replace the blue line with a black dashed line), 10 (use colors for symbols and lines), 11, and 12.

10 It is very good suggestions! We have numbered plots in Figure 1, and changed the color in Figures 6, 10, 11 and 12.

4. Page 7, line 3. There are other possible reasons accounting for the lower SOA yield. For example, this study uses higher temperature than the two previous studies mentioned.

We have added the following one sentence: In addition, the temperature in this study is higher than the previous studies, which

15 may be another reason accounting for the lower SOA yields in this work.

5. Page 8, line 5. Is this based on MCM simulation? Please add clarification in the text.

Yes, it is based on MCM simulation, to clarify it, we have added the following words to the original sentence: based on MCM simulation.

**20**

6. Page 13, line 17. "MPAN can be oxidized by OH to form 2-methyltetrols" is wrong. The products are 2-methylglyceric acid and related oligomers. This is repeated by later sentences in this paragraph. The authors should consider consolidate/rewrite this paragraph.

We have rewritten the paragraph:

The yield of MACR is generally greater in isoprene-NO2 irradiations and isoprene-O3 systems than that in isoprene-H2O2 irradiations. MACR can react to form MPAN in the presence of NO2, which can be oxidized by OH to form SOA precursors of epoxides (e.g., HMML, MAE), such as in the Nguyen et al. (2011) work. Epoxides can further be oxidized to produce 2-MG and related oligomers (Surratt et al., 2010; Lin et al., 2013; Nguyen et al., 2015). 2-MG-derived oligomers can be enhanced under lower RH (Zhang et al., 2011). Both the results from Nguyen et al. (2014) and MCM simulations further show that if there are enough OH radicals, most of MPAN can be further oxidized by OH to produce epoxides. However, since there were no extra OH sources in our systems, MCM simulations show that only 12% (24%) of MPAN under dry (humid) conditions was oxidized by OH to produce HMML and MAE. The maximum concentrations of HMML and MAE were only 6.8 and 2.7 ppb under dry conditions (Figure 4), which is too small to explain the yields of SOA in isoprene-NO2 irradiations.

10

5

7. Page 14, line 6. "Thus, IEPOX is not the major contributor to SOA in isoprene-NO2 system. This clearly demonstrates that MPAN and IEPOX related reactions were not dominant pathways for SOA formation in our isoprene-NO2 irradiations". The logic in this sentence is problematic. It cannot suggest MPAN reactions are not important in isoprene-NO2 irradiations. Again, a figure showing simulated MPAN, SCI, and a few other important products is needed at least.

- 15 The logic of the original sentence is indeed problematic. Following the reviewer's comments, the original sentence "This clearly demonstrates that MPAN and IEPOX related reactions were not dominant pathways for SOA formation in our isoprene-NO2 irradiations." has been deleted. As our response to main comment 2, we believe that MPAN has a small contribution to SOA in our reaction system because the reacted MPAN is low under our experimental conditions. In addition, based on data from FTIR and MS spectra we suggest that SCI-based oligomers are major components of SOA in isoprene-NO2 irradiations.
- 20 We have added a figure (Figure 13) to show the concentrations of important SOA precursors under our experimental conditions.

Taking the reviewer's comment, we have added the following sentences on page 18:

Lin et al. (2013) has reported that C4H6O3 was from MAE in MACR-NOx irradiations. Thus, C4H6O3 is probably formed from

8. Page 18, line 9. C4H6O3 is not formed from dehydration of 2-MG, but MAE (Lin et al., 2013 PNAS).

dehydration of 2-MG and MAE in oligomers. Considering the low yield of MAE in our system, we considered that most of  $C_4H_6O_3$ - based oligomers are probably contributed by 2-MG in our work.

5

10

15

20

**Response to Reviewer 2**

We greatly appreciate the time and effort that reviewer 2 spent in reviewing our manuscript. The comments are really thoughtful and helpful to improve the quality of our paper. Reviewer 2 has provided both major comments and other specific comments. Below we make a point-by-point response to these comments. According to editor's requirement, the response to the referee is structured in the following sequence: (1) comments from the referee in black color, (2) our response in blue color, and (3) our changes in the revised manuscript in red color.

The manuscript is comprehensive and provides useful information about the chemical composition of SOA formed from various pathways. However, there are a few points that the author should consider when making these arguments, and these issues should be addressed before the article is published in ACP.

**Major Comments**

First, the author draws the conclusion that RH does not have an effect on isoprene SOA yield under the OH oxidation pathway. However, Gaston et al. (2014) clearly show that RH has an effect on isoprene epoxydiols (IEPOX) reactive uptake into the acidic sulfate particles due to the dilution effect at higher RH. Because IEPOX is one of the main oxidation products of isoprene under OH pathway, the yield should show a negative correlation with RH. But the author's experimental results seem to not

- agree with those of past studies. I believe the main reason for non-RH dependence observed in this study is that the acidity of the  $H_2SO_4$  particles generated by this study is too low, resulting in RH not having an additional acidity dilution effect. The author should at least mention the discrepancy between this work and other studies, and point out limitations of this study. For
- 25 instance, page 7, line 19-20: the author made a conclusion that RH had little effect on the yield of isoprene with OH reactions.

Please note that this is probably true for generating self-nucleated isoprene SOA, but may not be true for isoprene SOA that are formed on acidic sulfate seed particles, as shown by a few other studies (Gaston, Riedel et al. 2014, Riedel, Lin et al. 2015, Zhang, Chen et al. 2018). These studies suggest that RH has an important effect on the formation of isoprene OH-heterogeneous generated SOA, due to the change of RH affecting the acidity of the seed particles. The author should include the literature and clarify the difference of RH effects on experiments because the seed particles in this study probably do not have enough

5

and clarify the difference of RH effects on experiments because the seed particles in this study probably do not have enough concentration to undergo full heterogeneous reactions.

We agree with the reviewer that there is discrepancy between our work and other studies. The initial burst of new particles happened at 10 min after the start of irradiations in isoprene- $H_2O_2$  system in our work and maximum number concentration was over 104. These new particles were considered to be  $H_2SO_4$  particles. Both theoretical simulation and experiments have

- 10 confirmed the formation of these new particles. We ever did the experiment. If the  $SO_2$  concentration in background air was removed to be below 50 ppt, the number concentration of  $H_2SO_4$  would be below 1000 /cm3. In addition, using the extrapurified background air, the mass concentration of SOA would be greatly reduced by one order of magnitude in isoprene-OH system. We have added a curve in the revised Figure 2 to show the variation of number concentrations of particles with time from the experiment of isoprene-H2O2, indicating the formation of H2SO4 and heterogeneous generated SOA. As the reviewer
- 15 pointed out, the mass concentration of H2SO4 was much smaller as compared to previous studies. Although the heterogeneous formation of SOA occurred in our system, compared with other studies the acid dilution effect was not obvious under higher RH condition. To make it clearer, we have added the following sentences on page 7 and references to clarify the differences between our work and previous studies.

---

## Author Response (AR2)

Dear Prof. Surratt,

Thank you very much for your kind message dated 28 Apr 2018 on our manuscript (acp-2017-1064). We have revised the manuscript in accordance with your advices and the comments of Reviewer 2. The response to the reviewer is included at the end of this message. In addition, we also carefully read through the manuscript to correct typographical, grammatical, and bibliographical errors. According to editor's requirement, the changes in our manuscript are displayed in different colors, including the addition in red color and the deletion in blue color.

Sincerely yours,

Yongfu XU

**Response to Reviewer 2**

We greatly appreciate the time and effort that reviewer 2 spent in reviewing our manuscript. Below we make a point-by-point response to these comments.

The revised paper reads well and is suitable to be published on APC. Just a couple of minor corrections: Page 8, line 15, please change has to had.

We have corrected it.

Page 11, line 18 and 20, oxford comma is recommended.

We have corrected it.

[revised manuscript text omitted]
_4H_6O_2$ as Kendrick base. Species separated by $C_4H_6O_2$ groups fall on the horizontal lines.

[Figure]

**Figure 13: MCM-simulated time profiles of SOA precursors in isoprene-NO₂ irradiations. SCI-derived oligomers were from the reactions of SCI with glyoxylic acid and ACETOL (solid lines for dry conditions, dashed lines for humid conditions).**

[Figure]

**Figure 14: Wall losses vs gas-phase reactions between H₂O and SCI**

**Figure 15: Mechanisms for SOA formation from the O₃ and OH oxidation channels of isoprene.**

Table 1 Expermental conditions of toluene and isoprene irradiations

| VOCs | # | T /K | RH /% | VOC /ppm | NO /ppb | NO₂ /ppb | Aim |
|---|---|---|---|---|---|---|---|
| Toluene | 1 | 304 | 6 | 0.915 | 4.3 | 307.7 | SOA size & yield |
| | 2 | 304 | 85 | 0.804 | 5.3 | 303.6 | |
| | 3 | 304 | 84 | 0.933 | 1.5 | 293.9 | |
| | 4 | 304 | 6 | 1.037 | 0.2 | 323.3 | |
| | 5 | 302 | 6 | 0.879 | 12.0 | 328.0 | |
| | 6 | 303 | 10 | 0.917 | 1.9 | 326.7 | FTIR |
| | 7 | 303 | 81 | 0.846 | 7.8 | 301.0 | |
| | 8 | 304 | 7 | 0.930 | 9.0 | 325.0 | FTIR with NaCl seeds |
| | 9 | 303 | 81 | 0.906 | 10.0 | 334.0 | |
| | 10 | 304 | 9 | 0.914 | 10.6 | 386.5 | UV/Vis |

| | | | | | | |
|---|---|---|---|---|---|---|
| 11 | 304 | 80 | 0.910 | 7.7 | 364.1 | |
| 12 | 305 | 79 | 0.927 | 12.1 | 288.1 | LWC by FTIR |
| 13 | 305 | 79 | 0.918 | 10.6 | 294.7 | LWC by SMPS |
| 14 | 302 | 7 | 0.896 | 7.0 | 353.0 | |
| 15 | 302 | 85 | 0.804 | 6.0 | 364.0 | size and |
| 16 | 301 | 7 | 0.850 | 0.0 | 311.7 | yield |
| 17 | 302 | 80 | 0.844 | 0.3 | 308.5 | |
| 18 | 303 | 7 | 0.901 | 0.0 | 299.5 | |
| 19 | 303 | 81 | 0.799 | 0.3 | 270.2 | FTIR |
| 20 | 302 | 80 | 0.828 | 0.2 | 273.0 | |
| 21 | 301 | 8 | 0.790 | 0.0 | 283.1 | |
| 22 | 303 | 9 | 0.873 | 3.0 | 301.0 | FTIR with |
| 23 | 303 | 79 | 0.827 | 4.0 | 325.0 | NaCl seeds |
| 24 | 303 | 8 | 0.823 | 0.1 | 332.5 | UV/Vis |
| 25 | 303 | 81 | 0.877 | 0.3 | 363.0 | |
| 26 | 305 | 81 | 0.831 | 1.5 | 288.8 | LWC by FTIR |
| 27 | 304 | 78 | 0.823 | 0.5 | 313.5 | LWC by SMPS |
| 28 | 303 | 7 | 0.810 | 0.2 | 295.1 | ESI-HRMS |
| 29 | 303 | 85 | 0.804 | 0.5 | 290.2 | ESI-HRMS |

(Isoprene — row label spanning rows 14–29)

**Table 2 List of major base units and their corresponding ratios of oligomers to total mass from SOA in isoprene-$NO_2$ irradiations.**

| Base unit | Unit name | SCI yields by MCM | Ratio of oligomers to total mass | | |
|---|---|---|---|---|---|
| | | | Dry /% | Humid /% | Delta /% |
| $CH_2O_2$ | $CH_2OO$ | 50.1 | 46.2 | 29.4 | 36.3 |
| $C_4H_6O_2$ | MACROO & MVKOO | 30.6 | 39.7 | 17.2 | 56.7 |
| $C_3H_4O_3$ | MGLOO | 11.3 | 19.1 | 3.7 | 80.4 |
| $C_2H_2O_3$ | GLYOO | 2.6 | 3.2 | 0.6 | 80.1 |
| $C_4H_6O_3$ | dehydrated 2-MG (or $CH_3OC_3H_3OO$) | (1.4) | 25.3 | 9.0 | 64.6 |
| $CH_2$ | - | - | 76.4 | 62.7 | 17.9 |
| $CH_2O$ | - | - | 78.5 | 67.6 | 13.8 |